# Negation Detection on Mexican Spanish Tweets: The T-MexNeg Corpus

Gemma Bel-Enguix [1],[†] , Helena Gómez-Adorno [2],[*],[†] , Alejandro Pimentel [1] and Sergio-Luis Ojeda-Trueba [1] and Brian Aguilar-Vizuet [3]

1   Instituto de Ingeniería, Universidad Nacional Autónoma de México, Circuito Escolar, Ingeniería S/N, C.U., Coyoacán, 04510 Ciudad de México, Mexico; gbele@iingen.unam.mx (G.B.-E.); pimentel@comunidad.unam.mx (A.P.); sojedat@iingen.unam.mx (S.-L.O.-T.)
2   Instituto de Investigaciones en Matemáticas Aplicadas y en Sistemas, Universidad Nacional Autónoma de México, Circuto Escolar 3000, C.U., Coyoacán, 04510 Ciudad de México, México
3   Facultad de Ciencias, Universidad Nacional Autónoma de México, Circuito Exterior, S/N, C.U., Coyoacán, 04510 Ciudad de México, México; brian1297av@ciencias.unam.mx
*   Correspondence: helena.gomez@iimas.unam.mx
†   These authors contributed equally to this work.

**Abstract:** In this paper, we introduce the T-MexNeg corpus of Tweets written in Mexican Spanish. It consists of 13,704 Tweets, of which 4895 contain negation structures. We performed an analysis of negation statements embedded in the language employed on social media. This research paper aims to present the annotation guidelines along with a novel resource targeted at the negation detection task. The corpus was manually annotated with labels of negation cue, scope, and, event. We report the analysis of the inter-annotator agreement for all the components of the negation structure. This resource is freely available. Furthermore, we performed various experiments to automatically identify negation using the T-MexNeg corpus and the SFU Review$_{SP}$-NEG for training a machine learning algorithm. By comparing two different methodologies, one based on a dictionary and the other based on the Conditional Random Fields algorithm, we found that the results of negation identification on Twitter are lower when the model is trained on the SFU Review$_{SP}$-NEG Corpus. Therefore, this paper shows the importance of having resources built specifically to deal with social media language.

**Keywords:** negation; machine learning; Mexican Spanish; Twitter





## 1. Introduction

Negation is a complex phenomenon of language that shows a wide range of variation, especially in what can be called Netspeak [1] or Computer-Mediated Communication (CMC) [2]—in Spanish, 'comunicación tecleada' [3]. Therefore, to have a more complete understanding of how negation works in the language of the Internet, it is necessary the use of corpora extracted from social media platforms that specifically annotate and deal with this phenomenon. Detecting, interpreting, and knowing different mechanisms of negation on social media is a key issue for a computational approach to automatically find opinions and sentiments of the users. Thus, this research is relevant because it has an impact on the study of marketing strategies, voting intention, people's mood, etc [4].

Twitter is a microblogging service that illustrates the features of netspeak; it is open, and its use is widely spread across all types of communities. Therefore, building and studying corpora based on Twitter is a very convenient strategy to study the traits of netspeak and, more specifically, negation.

Reitan et al. [5] built a corpus of negation in English containing twitter messages, focused on sentiment analysis. However, in general there is a lack of resources to approach this topic, even in English.

Additionally, Spanish is a largely spoken language, with native speakers in Europe and America. Mexico has the largest Spanish speaker community, with 123 million in

2020 [6] and 9.5 million users on Twitter (https://es.statista.com/estadisticas/1172236/numero-de-usuarios-activos-mensuales-twitter-mexico-sistema-operativo; accessed on 12 February 2021). This causes dialect diversification, not only in the lexicon but also in morphology, syntax, and several slang expressions. Therefore, the language on the Internet [2], WhatsApp, Twitter, Facebook, etc., also shows great diversity. Because of this, it is crucial to have a corpus of reference of negation in Mexican Spanish based on social media language.

However, to the best of our knowledge, there are no Spanish corpora that are annotated with negation in Twitter in Spanish [7]. This paper presents T-MexNeg, the first corpus annotated with negation in Twitter in Mexican Spanish. Although the main objective of this paper is not to study the language of the Internet, this work can be a solid contribution to the topic, providing data for future studies.

The present work is structured as follows. Section 2 is devoted to explaining some related literature involving corpora of negation in English and Spanish. Section 3 introduces our corpus of Mexican Spanish Tweets and the annotation protocol, explaining the main criteria for annotation, and the specific tags we adopted. The obtained resource is freely available and can be used in different natural language processing pipelines (https://gitlab.com/gil.iingen/negation_twitter_mexican_spanish; accessed on 12 February 2021). We explain the design of the experiments to automatically identify negation in Section 4 and the results of the annotation and the experiments in Section 5. Section 6 discusses the results and the challenges that we faced during annotation. The paper concludes with final remarks and future work in Section 7.

## 2. Related Work

Several negation scope corpora are described in the literature for different domains and languages. The development of such a resource will enable the training of supervised machine learning methods for detecting negation cues, events, and scopes. Next, we describe some of the more relevant negation resources existing in the literature.

Most of the available negation corpora are in English. Although there is not a standard annotation criterion, the tags used in such corpora are: (a) negation cue; (b) scope; (c) focus; and (d) event.

### 2.1. English Corpora

Table 1 summarizes the most relevant corpora annotated with negations in the English language. Most of the English negation corpora belong to the medical domain and consist of abstracts or papers from biomedical research, clinical text, and electronic health records. The first corpus marked with negation is the BioInfer corpus [8]. It has 1100 sentences, of which 163 are tagged with negation cue and scope. Another corpus in the medical domain is BioScope [9]. They annotated negation and speculative keywords along with their scope in 20,924 sentences that belong to three different sources: medical free texts, biological full papers, and biological scientific abstracts. The BioScope corpus has 2720 sentences marked with negation. The last two corpora in this domain are NegDDI-DrugBank [10] and NEG-DrugDDI [11]. Both are annotated only with the scope tag. NegDDI-DrugBank [10] has 5806 sentences, with 1399 sentences marked with negations. The NegDDI-DrugBank [11] has 6648 sentences, with 1448 marked with negation.

In the journal domain, the corpora usually contain news articles, but they can also have opinion articles or comments about the news. The PropBank Focus [12] is the first negation corpora that mark the negation focus. They marked 3993 verbal negation contained in 3779 negative sentences. Meanwhile, the SFU Opinion and Comments Corpus [13] provides 1043 comments annotated with 1397 negation keywords, 1349 instances of scope, and 1480 instances of focus. In the domain of reviews, corpora contain user opinions about any type of product (like books, phones, movies, etc.). The Product Review [14] corpus belongs to this domain, consisting of product reviews extracted from Google Product Search. It contains 679 sentences annotated with negation from a total of 2111 sentences.

Another review corpus is the SFU Review$_{EN}$-NEG Corpus [15] (one of the best-known resources in this field). It consists of 400 reviews obtained from the Epinions.com webpage, with a total of 17263 sentences, of which 3107 contain at least one negation structure. In the literature domain, a corpus annotated with negation is the ConanDoyle-neg [16], and it is composed of the Conan Doyle stories: The Hound of the Baskervilles and The Adventure of Wisteria Lodge. It is the first English corpus that tagged event and scope. It has a total of 4423 sentences, 995 with negation. Finally, another unique corpus is Deep Tutor Negation corpus [17] which consists of texted dialogues from students interacting with an Intelligent Tutoring System. It contains 27785 responses, 2603 marked with negation. This corpus can be included in the educational domain.

**Table 1.** English corpora annotated with negation.

| Corpus | Year | Domain | Scope | Event | Number of Sentences | Annotated Negations |
|---|---|---|---|---|---|---|
| BioInfer | 2007 | Medical | Yes | No | 1100 | 163 |
| BioScope | 2010 | Medical | Yes | No | 20,924 | 2720 |
| NegDDI-DrugBank | 2013 | Medical | Yes | No | 5806 | 1399 |
| NEG-DrugDDI | 2014 | Medical | Yes | No | 6648 | 1448 |
| PropBank | 2011 | Journal | No | No | 3779 | 3993 |
| SFU Opinion and Comments | 2019 | Journal | Yes | No | 1043 (comments) | 1397(negation cues) |
| Product Review | 2010 | Review | No | No | 2111 | 679 |
| SFU Review$_{EN}$-NEG | 2012 | Review | Yes | No | 17,263 | 3017 |
| Conan Doyle-neg | 2012 | Literature | Yes | Yes | 4423 | 995 |
| Deep Tutor Negation | 2016 | Educational | Yes | No | 27,785(response) | 2603 (responses) |

*2.2. Spanish Corpora*

Among the Spanish bibliography related to negation annotations, many of them deal with negation in different areas. Table 2 summarizes the Spanish corpora annotated with negation. The first Spanish Corpus annotated with negation in the domain of reviews is the SFU Review$_{SP}$-NEG [18] which consists of 400 reviews extracted from the Ciao.es website. It includes a total of 9455 sentences, with 3022 sentences containing at least one negation structure. This corpus is the Spanish version of SFU Review$_{EN}$-NEG Corpus [15]. In addition, the SFU Review $_{SP}$-NEG [18] is the first Spanish corpus with negation annotations that includes a detailed description of the typology of negation patterns in Spanish, with event and scope annotated.

There are other Spanish Corpora that deal with negation in different domains: The UAM Spanish Treebank corpus [19] contains 1501 sentences from the newspapers El País and Compra Maestra. In total, 160 sentences were identified with negation and syntactically annotated. Similar to the UAM Spanish Treebank corpus [19], there is the NewsCom corpus [20], but, instead of news, the corpus consists of 2955 comments posted in response to 18 news articles from online Spanish newspapers. The news articles cover a variety of nine different topics, two articles per topic. The corpus consists of 4980 sentences, 2247 marked with negation. It is annotated with focus, cue, and scope. In the medical domain, the IxaMed-GS corpus [21] includes 75 electronic health reports with 5410 sentences from the Galdakao Unsansolo Hospital in Spain. In total, 763 entities are syntactically and semantically annotated with negation. The UHU-HUVR corpus [22] consists of 604 clinical reports with 8412 sentences, 1079 with annotated negations from the *Virgen del Rocío* Hospital in Spain.

In addition, the IULA Spanish Clinical corpus is in the medical domain with clinical reports [23] from a hospital in Barcelona. It contains 300 clinical reports with 3194 sentences, being 1093 tagged with negation cues. In this work, the authors created their own guidelines to elaborate the corpus. Another clinical negation corpus in Spanish is the NUBES corpus [24] with 29,682 sentences from anonymized health records sup-

plied by a Spanish private hospital. It contains 7567 sentences with negation, annotated with scope, event, and cue. Most of the corpora cited here were described in detail by Jiménez-Zafra et al. [4].

**Table 2.** Spanish corpora annotated with negation.

| Corpus | Year | Domain | Scope | Event | Number of Sentences | Annotated Negations |
|--------|------|--------|-------|-------|---------------------|---------------------|
| SFU Review$_{SP}$-NEG | 2018 | Reviews | Yes | Yes | 9455 | 3022 |
| UAM Spanish Treebank | 2013 | Newspaper | Yes | No | 1501 | 160 |
| NewsCom | 2020 | Comments | Yes | Yes | 4980 | 2247 |
| IxadMed-GS | 2015 | Medical | No | Yes | 5410 | 763 |
| UHU-HUVR | 2017 | Medical | Yes | Yes | 8412 | 1079 |
| IULA Spanish Clinical report | 2017 | Medical | Yes | No | 3194 | 1093 |
| NUBES | 2020 | Medical | Yes | Yes | 29,682 | 7567 |

*2.3. Methods and Shared Task for Negation Identification*

The development of systems that automatically detect negation is a well-known natural language processing task. This task is usually divided into four fundamental assignments: (I) negation cue detection; (II) negation event recognition; (III) negation scope detection; and (IV) focus detection [25]. There are essentially three different approaches applied to the development of these systems.

First, rule-based systems (NegEx, ConText, DEEPEN, NegMiner, etc.) emerged from the need to automatically extract information from clinical records. The NegEx system, developed by Chapman et al. [26], is one of the most popular systems, and it uses a list of triggers that indicate the presence of negation. Later, other techniques emerged such as ConText [27], DEEPEN [28], and NegMiner [29]. Costumero et al. [30] adapted and translated NegEx into Spanish and used it to detect negation in clinical texts.

Second, machine learning techniques have had exponential growth in the last decade. Agarwal and Yu [31] developed a classifier based on a Conditional Random Field algorithm to automatically identify negation and its scope in biological and clinical text. Li et al. [32] proposed a semantic parsing approach to learn the scope of negation. Cruz Díaz et al. [33] implemented a Naive Bayes classifier that identifies negation and speculation signals and their scope in clinical texts. Jiménez-Zafra et al. [34] presented a machine learning system based on a Conditional Random Field that processes negation in Spanish. They focused on two tasks, negation cue detection, and scope identification, outperforming state-of-the-art results for cue detection and being the first system that performs the task of scope detection for Spanish.

Third, the most recent approaches suggest the use of deep learning techniques. Fancellu et al. [35] used neural network architectures to automatically detect negation scope. Qian et al. [36] proposed a Convolutional Neural Network-based model to address speculation and negation scope detection. Recently, Khandelwal and Britto [37] utilized transformer-based architectures such as BERT, XLNet, and RoBERTa to detect negation and speculation.

It is important to highlight that none of the previous works on negation detection explore the domain of social media. Given that social media is nowadays one of the main communication platforms and attracts research on different natural language processing tasks, it is important to have corpora for exploring automatic methods for negation detection.

In another vein, some works deal with negation on social media in the context of sentiment analysis. Two key works in this line of research were performed by Mohammad et al. [38] and Wiegand et al. [39]. The former labels with the "_NEG" suffix every word from the negation cue to a punctuation mark and shows how this strategy improves the performance of the system for detecting polarity. The latter approaches several methods

for approaching modeling of negation in sentiment analysis. However, these works are mainly focused on the detection of polarity and do not tackle the structure of negation.

Several shared tasks have emerged to tackle NLP problems regarding negation. BioNLP'09 Shared Task 3 [40] centered on the detection of negations and speculation statements concerning extracted events based on the GENIA event corpus of biomedical abstracts written in English. i2b2 NLP Challenge [41] focused on the automatic extraction of concepts, assertions, negation, and uncertainty on reports written in English provided by the University of Pittsburgh Medical Center. *SEM2012 Shared Task [42] was dedicated to the identification of the negation, its scope, and focus. Two datasets were produced for the task, the CD-SCO corpus of Conan Doyle stories annotated with scopes and the PB-FOC corpus, which provides focus annotation on top of PropBank, both written in English. ShARe/CLEF eHealth Evaluation Lab 2014 Task 2 [43] focused on facilitating understanding of information in clinical reports by extracting several attributes such as negation, uncertainty, subjects, severity, etc. In this task, the MIMIC II Dataset of clinical reports written in English was used. More recent shared tasks, such as NEGES 2018 [44] and NEGES 2019 [45], aim to advance the study of computational linguistics in Spanish, proposing subtasks including annotation guidelines, negation cues detection, sentiment analysis, and the role of negation in sentiment analysis. NEGES uses the SFU Review$_{SP}$-NEG corpus of reviews written in Spanish.

## 3. Corpus of Negations in Mexican Spanish: The T-MexNeg

We collected tweets from Mexican users from September 2017 to April 2019. These tweets were extracted by consuming data from the Standard streaming API. We received every tweet offered by Twitter with the language tag "es"; furthermore, we filtered the tweets by user's location field in search of the "mx" tag to limit the collection to the Mexican territory, no other filtering was considered for the extraction process. To elaborate our corpus, we took a full random subset of tweets from this collection.

In general, our corpus follows the annotation protocol established by Jiménez-Zafra et al. [18]. However, several modifications were made to adapt the tag-set to the special features of our tweets: short messages, external references, hashtags, netspeak traits, Mexican language structures, Mexican slang, typos, abbreviations, and other problems that make it very difficult to decide the extent and modality of negations in the corpus.

The corpus has no pre-processing of tweets. We did not change the data or perform any spelling corrections. Besides, it contains only tweets with some textual information. Those that had only audiovisual content or emojis were not extracted. However, we did not leave out tweets that had syntactic negations along with emojis, audiovisual content, mentions, or hashtags. Sometimes, such content gave us additional or substantial information to understand and study the negation in the tweet, and leaving them out is a loss. Finally, we also included tweets that were answers or retweets, i.e., the type and form of the tweet did not matter to the extraction and annotation process.

### 3.1. Annotation Protocol

Our first step towards negation annotation is the classification of tweets, whether there is a negation present at all to analyze or not. As explained above, we sampled a collection of tweets and carried out a manual binary classification; 4895 tweets were found to have some sort of negation in their content. This preliminary step allowed us to carry out a much more detailed analysis of the negations within the previously detected tweets. To achieve this, an independent three-way tagging was carried out to identify each particular syntactic negation as well as their respective components.

In Spanish, there are three basics levels of negation: lexical, morphological, and syntactic. This work only approaches the study of syntactic negation. According to RAE [46], negative sentences express false states or the nonexistence of the action that is in the sentence. Syntax negation is a syntax operator word that affects the whole sentence or a section of it. This syntax operator is called negation cue. They can be adverbs, prepositions, indefinite

pronouns, and conjunctions. Usually, in Spanish, negation cues precede the verb, but they can also appear postponed.

The section affected by the negation cue is called scope. The words that are specifically reached by it, which can be verbs, nouns, or phrases, are referred to as event [47]. Therefore, the basic requirements to create a negative sentence are the negation cue, the scope, and the event. Figure 1 presents three sentences with negation cue annotated in red, negation event annotated in purple, and negation scope in blue. In the first sentence "*@072CDMX Esperemos que le den solución lleva más de 24 horas y no la atienden*" (@072CDMX Let's hope they give you a solution it takes more than 24 h and they don't attend her) we can observe that the negation cue is the word "*no*" (don't), the negation event is "*la atienden*" (attend her), and the negation scope is "*no la atienden*" (don't attend her).

**Figure 1.** Example of tagged tweets with cue, event, and scope.

Our annotation protocol follows the criteria suggested by Martí et al. [47], but includes some modifications caused by the specific features of a Twitter-based corpus. In general, we identify three main negation components: Negation Cue, Event, and Scope. We also differentiate among three types of negation cues: Simple Negation (Neg_exp), Related Negation (Neg_rel), and False Negation (No_neg). In the next sections, we explain the details of each category. In addition, we simplified the categorization because the tweets are shorter than the reviews analyzed by the followed criteria. For example, we use the tag Neg_rel when two negation cues appear in the same sentence but there is only one event negated.

### 3.1.1. Negation Cues

The negation cue tags are used to identify the negations that appear in a tweet. The difference between Simple Negation and Related Negation lies in the syntactic distribution of the cues because there are several negative cues that are coordinated. The difference between Simple Negation and False negation has a semantic and pragmatic nature. Some negative cues simply do not negate anything.

**Simple Negation (Neg_exp)**

It refers to the negation cues that are not linked to other negation cues (1). Thus, the Scope and the Event are only directly related to this negation. To identify this, we created the tag neg_exp.

(1)    **No** tengo ganas de ir al baño ahora.
       *I do not feel like going to the bathroom right now*

**Related Negation (Neg_rel)**

This label is used for negation cues that are linked to other negation cues in the sentence (2) and are dependent on them. The related negation does not have an event or scope and it is part of the scope of the main negation. An special case is the use of *ni...ni* (neither, nor) without a new event (3).

Martí et al. [47] proposed in their work the complex negation tag that is composed of negation reinforcement, the negation with modifiers, comparative negations, and negation phrases. We decided to simplify these tags by identifying the relationship between the principal negation cue and the related cue. We privilege the fact that the second negation in a sentence is related to the first one, regardless

of the meaning. In the next two examples, we boldfaced only the Related Negation (Neg_rel) tag.

(2)    No me ha gustado **nada** la película.
        *I didn't like the movie at all.*

(3)    No quiero **ni** pollo **ni** pan.
        *I want neither chicken nor bread.*

**False negation (No_neg)**

This tag is used with negation cues that do not negate anything at a semantic level, as well as with some abbreviations such as *no./no*—which are used to represent the word *number* in Spanish. Quite frequent idiomatic phrases that are discursive markers, rather than negations, belong to this category. Some examples are: *no mames* (no way) (5), *de nada* (you are welcome), and *nada más* (just/only) (4). Martí et al. [47] mentioned that there exist both complex and simple structures with negation cues that do not express semantic negations. In our annotation protocol, both structures are tagged in the same way.

(4)    **Nada más** quiero darte un beso.
        *I just want to kiss you.*

(5)    **NO MEMESSS**, no hay papel.
        *No way, there is no toilet paper*

3.1.2. Negation Event

The Event labels the word or words that are specifically negated. The Events are usually verbs because most of the negation cues are adverbs: *no, jamás, tampoco* (no, never, neither/either). In verbal periphrasis and other types of verboids (infinitive, gerund, and participle), we tagged the auxiliary and the principal verb as the Event. The same tag is used for the clitics and the other adverbs or words (that are not negation cues) that were related to the negated verb (6). Sometimes the Event is a noun or idiomatic phrase; usually, this happens with a preposition such as *sin* (7) (without) or with *no* plus a prepositional phrase (8).

(6)    No **la vi** ayer.
        *I did not see her yesterday.*

(7)    La vida es la misma sin **ti**.
        *Life is the same without you.*

(8)    NO **A LA COMPRA DE VOTOS**.
        *NO TO VOTES BUYING.*

3.1.3. Negation Scope

This tag corresponds to all words that are affected by the negation. The scope includes: (a) the negation cues, except the ones that are considered no_neg; (b) the Event; and (c) all the words that are affected by the negation such as adjectives or nouns (9). In the case of the subordinated clause with negation, the main sentence is not taken as part of the negation Scope (10).

(9)    Yo **no seré una de las muchas del montón**.
        *I would not be like the rest of the girls.*

(10)   Quiero saber que **no me engañastes**.
        *I want to know that you did not cheat on me.*

*3.2. Annotation Methodology*

Based on the description of the above annotation tags, we created an annotation guide that specifies how the tweets should be labeled and under which criteria (the annota-

tion guidelines are publicly available at: https://gitlab.com/gil.iingen/negation_twitter_
mexican_spanish/-/blob/master/annotation_guidelines.pdf; accessed on 12 February
2021). We performed a manual tagging process for labeling each of the tweets in the
corpus. The tagging process involved three teams of two annotators. All annotators were
linguistics students, and we had a chief linguist who helped to solve cases where the
linguist annotators disagreed. Each team tagged the full dataset using the Dataturks (
https://dataturks.com/; accessed on 12 February 2021) online platform, which allows this
type of annotation for computational purposes. Thus, for each tweet, we obtained three
different annotations.

To facilitate the handling of the tagged corpus, we applied a transformation of the
original JSON file that was obtained from Dataturks to XML. The general structure to
which the tagged corpus was transformed is shown in Figure 2. The full corpus is freely
available and can be used in different natural language processing pipelines (https://gitlab.
com/gil.iingen/negation_twitter_mexican_spanish; accessed on 12 February 2021).

```
<tweet>
    <content>
        <neg_structure>
            <scope>
                <negexp class='simple/related/no_neg'>
                </negexp>
                <event>
                </event>
            </scope>
        </neg_structure>
    </content>
</tweet>
```

**Figure 2.** General structure of the negation tagging.

With the tweets already tagged, we computed the statistics regarding the word counts
for each tweet in the corpus. Table 3 shows the general statistics for both types of tweets,
with and without negation; therefore, we can observe that, in both cases, tweets have
a minimum of 1 word and a maximum of 29 words, so that independent of presence
or absence of negation, tweets in our corpus range from 1 to 29 words. We can also
observe in the table that tweets that contain negations tend to be almost three words
longer on average than a regular tweet, however, the standard deviation of the number
of tokens is quite high for the length to be a reliable distinctive feature. We also reviewed
which were the most used tags and which types of negation cues are the most common.
Table 4 shows the negation cue tags distribution (Freq) along with some statistics such
as the average, minimum, and the maximum number of words in each tag. As can be
seen, most of the negation cues are simple, while the number of related and false negation
(no_neg) are very similar. Table 4 also shows the frequency of the scope and event tag,
where it can be observed that the average number of words within the scope is larger than
in the rest of the tags. This variation is expected given that the scope corresponds to all
words affected by the negation. Concerning the event tag, it is typically composed of only
one word (average); however, there can be more than one word that is specifically negated.
Note that, since we did not perform any pre-processing of the tweets, there are several
variations of every word, including capitalization, emphasis by the repetition of a vowel,
and other phenomena. This can be seen in Appendixes A–C.

**Table 3.** General statistics computed from word counts on each tweet.

|  | With Negation | Without Negation | Full |
|---|---|---|---|
| Average number of words per tweet | 13.65 | 11.08 | 12.00 |
| Standard Deviation | 5.31 | 6.51 | 6.23 |
| Sample variance | 28.21 | 42.33 | 38.80 |
| Minimum number of words in a tweet | 1 | 1 | 1 |
| Maximum number of words in a tweet | 29 | 29 | 29 |
| Total number of words | 66,824 | 97,628 | 164,452 |
| Tweets count | 4895 | 8809 | 13,704 |

**Table 4.** Frequency distribution by negation tags. Right-side data are expressed in number of words per tag.

| Tag | Freq | Average | Standard Deviation | Minimum | Maximum |
|---|---|---|---|---|---|
| Simple | 5307 | 1 | 0.39 | 1 | 4 |
| Related | 392 | 1 | 0.22 | 1 | 2 |
| No_neg | 324 | 2 | 0.71 | 1 | 4 |
| Scope | 5143 | 5 | 2.98 | 1 | 22 |
| Event | 5093 | 1 | 0.97 | 1 | 10 |

Figure 3 shows the 15 most frequent negation cues in the corpus. We highlight that the five most frequent negation cues represent 91% of the total tags: "no", "nunca", "nada", "ni", and "sin".

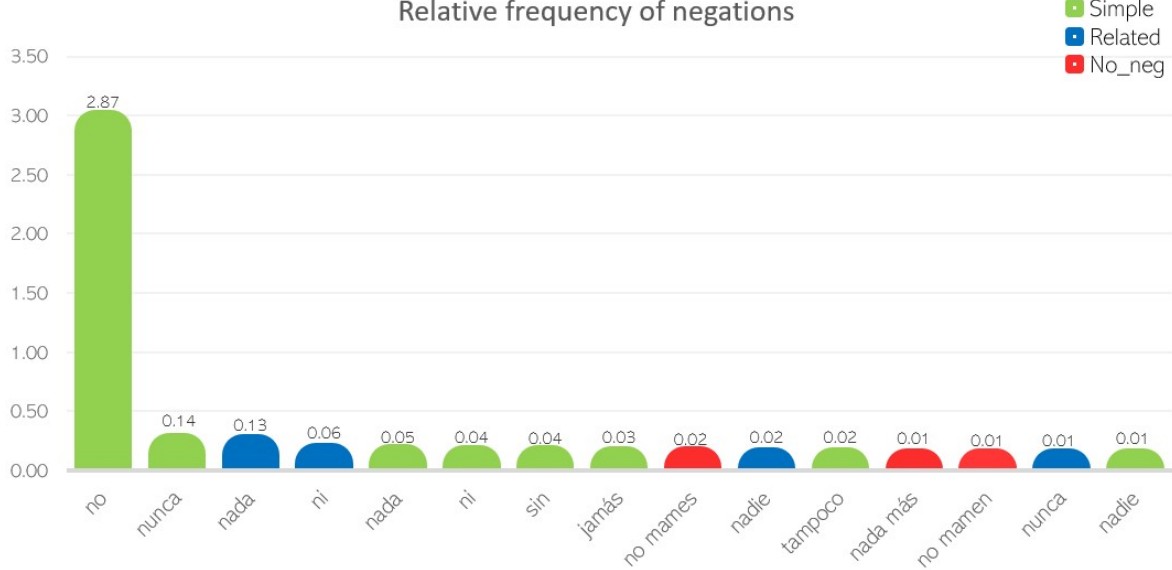

**Figure 3.** Histogram of negations in the corpus. Green stands for neg_ex, blue refers to neg_rel tag, and red means no_neg.

### 3.3. Challenges of the Annotation

In this section, we describe some issues we found related to the conditions of Twitter and the way users write on social media in general.

**Emphatic-pragmatic effects in repetitions:** One of the main problems in the annotation process of this corpus is related to the writing conditions of Twitter, and the impact that such conditions have on language. Twitter is a social network that reminds of orality because the users change their writing to communicate fast and expressively [3] (similar to on other instant message platforms). For example, some tweets have

more than one negation cue to indicate an emphatic-pragmatic effect, but with the same negation, event, and scope (11).

(11)     AAAAAAHHHHHHHHHHHHHHHHHH NO NO NO NO NO ME PUEDES HACER ESTO.
*AAAAAAHHHHHHHHHHHHHHHHHH NO NO NO NO NO YOU CAN'T DO THIS TO ME*

We solved this problem by tagging all the NOs that appear in the tweet as only one simple negation (Neg_exp) and only tagging one event: "ME PUEDES". There were just a few cases where the same thing happens with the event, i.e., the event is also repeated.

**Orthography** Frequently, the users try to represent a certain tone in the writing, so they modify their spelling or typography in a way that is more convenient for this. A good example of this is the word *ño* (12) (nope) that had to be included in the extraction list of negations so that it was considered in the final tagged corpus.

(12)     Ño quiero, ño quiero.
*Don't wanna, don't wanna.*

Misspellings cause other problems and confusions, such as the one between *sino* (but) and *si no* (if not/otherwise). Indeed, in many examples, it is not clear whether the meaning is former or the latter. Examples (13) and (14) show two different spellings for the same meaning, *si no*. We did not tag the expressions *sino* (but) or *si no* (if not) when they appear in the tweet, although this can bring some errors.

(13)     Es porque sino se me desordena mucho el TL.
*It is because otherwise TL gets very messy*

(14)     Si no me muero en estas dos semanas, no muero mas.
*If I don't die in these two weeks, I won't die anymore.*

**The Twitter Meta-language and Trends** Usually, certain trends that modify the prototypical form of the Spanish negations appear on Twitter. For example, the phrase *Dijo nadie nunca* (15) (no one ever said) was a popular tweet (trend) that works as an idiomatic/sarcastic phrase where the negation expression *nadie nunca* (no one ever) negates the verb *dijo* (said).

(15)     Uy, gracias por publicar tu horóscopo!! ya me pongo a leerlo. Dijo nadie, nunca.
*Oh, thanks for sharing your horoscope!! I'll read it right away. Said no one ever.*

Another popular phrase, *Ni Obama* (neither Obama) (16), is a new way to say *nadie* (nobody) on Mexican Spanish social media. Thus, for this case, we tagged all the phrase as a negation. We tagged *nadie nunca* and *Ni Obama* as a single negation cue because they only negate one event.

(16)     Esa motivación no la tiene ni Obama!
*Not even Obama has that motivation!*

Twitter has a specific writing style, such as hashtags. We decided not to tag the hashtags that are already a unity or a trending topic such as #*NoEraPenal* (#Itwasnotapenalty) (17). We only tagged the hashtag as a negation cue when the negation cue was the only word that a hashtag had (18).

(17)    Día 2 #NoEraPenal.
        *Day 2 #Itwasnotapenalty*

(18)    **#no** lo seré.
        *I will **#not** be.*

### 3.4. Results of the Annotation Process

Let us recall that the annotation process was carried out in two stages: a binary classification of negation presence and the full negation annotation process. As mentioned in Section 3.1, this was a search for negation over 13,704 tweets, from which 4895 tweets were found to have some kind of negation. Moreover, the annotators presented a Cohen's $\kappa$ agreement average of 0.897, a very high agreement indeed. Table 5 shows the percent of agreement and Cohen's $\kappa$ score for each pair of annotators. For the 4895 tweets, we took every tweet tagged with negation by at least one annotator.

**Table 5.** Agreement scores by each pair of annotators of the binary classification of negation.

| Annotator Pair | 1 & 2 | 1 & 3 | 2 & 3 |
|---|---|---|---|
| Percent of agreement | 96.477 | 96.065 | 97.569 |
| Cohen's $\kappa$ score | 0.890 | 0.875 | 0.926 |

As for the negation annotation, the three different tagging results mentioned in Section 3.1 were used to determine the multi-rater inter-annotator agreement and the most reliable tags for each tweet so that we could obtain a definitive tagged corpus.

For this process, the tagging results were aligned at the word level using basic whitespace tokenization. That is, each token represents a case to be tagged with one of four categories: negation, event, scope, or none. Due to the nature of the problem and the tagging process, it was decided to use Randolph's free-marginal multi-rater [48] kappa for three annotators with four categories. This analysis resulted in an overall agreement of 0.936 (0.84 when ignoring words without tags) and a 0.915 of free-marginal kappa agreement (0.787 when ignoring words without tags) which reflects a very high agreement inter-annotator agreement.

In addition to the above, the alignment allowed for the identification of those labels with the highest reliability. That is, by having three independent labels, in the cases where there was disagreement, we were able to search for agreement in two out of three to set the repeated label as the definitive one in the final corpus that is offered.

## 4. Experiments on Automatic Negation Identification

We evaluated two methodologies for automatic negation identification in the T-MexNeg corpus. The first methodology consists of a baseline system that only uses a dictionary to identify negation cues. The second methodology consists of a system based on a Conditional Random Field (CRF) [49] algorithm. This algorithm is used for the structured prediction of labels by considering the labels of the previous and following words in a sentence. To compare the results of the automatic negation identification, we performed the same experiments on the SFU Review$_{SP}$-NEG [18] corpus. In this way, we can compare the performance of methodologies in different variations of Spanish as well as in different domains.

To handle the information in the T-MexNeg corpus, a prior treatment on the data was executed. We used the spaCy's (https://spacy.io/, accessed on 30 January 2021) library for Natural Language Processing in Python to tokenize each tweet into words or tokens. Then, using the same library, we labeled the Part of Speech (POS) tag of each token. Finally, the labels of the negation cue, event, and scope given in both corpora were translated into a BIO [50] tagging scheme. In the case of the SFU Review$_{SP}$-NEG corpus, the text is already tokenized and labeled with POS tags, so such process was not necessary. We only arranged the data in the same way as in the T-MexNeg corpus.

The dictionary-based system consists of a classifier that uses a list of the most commonly used negations in the Spanish language as the only search argument. This system validates each word in a sentence; if a given word appears in the dictionary, then this word is labeled as a negation cue. We considered five of the most common negative adverbs in Spanish language: *no* (no), *nada* (nothing), *nunca* (never), *tampoco* (neither), and *jamás* (ever). The dictionary-based system obtained an F1-score of 0.91 on the T-MexNeg corpus, which indicates that, including only some of the most frequent labels, a high percentage of negation expressions is covered. In the next section, we cover in detail the obtained results.

To improve the results achieved by the dictionary-based system, we developed a system based on a Conditional Random Field algorithm [49]; this is a framework modeled as a probabilistic graph to segment and label sequence data. CRF offers several advantages over other models, including the ability to relax independence assumptions and better relate training variables, in other words, it takes context into account by making connections between neighboring nodes (words). Jiménez-Zafra et al. [34] implemented a CRF-based system and presented results of the state-of-the-art for the SFU Review$_{SP}$-NEG corpus where for cue identification the system outperforms state-of-the-art results, while for scope detection they provide the first experimental results. Agarwal and Yu [31] showed that CRF achieves high F1 scores and is accurate when identifying negation cues and its scope in clinical notes.

The feature set used in our study is inspired by the work done by Loharja et al. [51] for the shared task of the Workshop NEGES 2018 [44]. They implemented a supervised approach combining CRF and several features for negation cue detection in Spanish for training the model. This approach ranked first in the official testing results of the Workshop. Our feature set used to train the model includes information of the word under observation, as well as the prior and posterior word; the word itself in upper- and lowercase is added to the feature list. Basic characteristics of how the word is written are also added to the features, for example, whether the word is constructed by alphanumeric, only alphabet, or only numeric characters. We also identify if the word starts with an uppercase letter, the length of the word, and its part-of-speech tag. We can see the complete features listed below.

1.  *WORD.Lower*: The observed word in lowercase.
2.  *WORD.Upper*: The observed word in uppercase.
3.  *WORD.LastTwoChar*: The last two characters of the observed word.
4.  *WORD.LastThreeChar*: The last three characters of the observed word.
5.  *WORD.LastFourChar*: The last four characters of the observed word.
6.  *WORD.LastFiveChar*: The last five characters of the observed word.
7.  *WORD.IsUpper*: Is the observed word is uppercase?
8.  *WORD.IsTitle*: Does the observed word begin with a character in uppercase?
9.  *WORD.IsDigit*: Is the observed word a numerical character?
10. *POS*: POS Tag of the observed word.
11. *POS.FirstChar*: First two characters of the POS tag.
12. *WORD.IsAlnum*: Is the word constructed by alphanumeric characters?
13. *WORD.IsAlpha*: Is the word constructed only by alphabet characters?
14. *WORD.IsNum*: Is the word constructed only by numeric characters?
15. *WORD.IsLower*: Is the word constructed by lowercase characters?
16. *WORD.Lenght*: Length of the word.
17. *±1WORD.Lower*: The lowercase word prior and posterior to the observed word.
18. *±1WORD.IsTitle*: Does the word prior and posterior to the observed word starts with an uppercase character?
19. *±1WORD.IsUpper*: Is the word prior and posterior to the observed word uppercase?
20. *±1POS*: POS Tag of the word prior and posterior the observed word.
21. *±1POS.FirstChar*: First two characters of the POS tag of the word prior and posterior to the observed word.

The CRF-based system achieved an F1-score of 0.95 in the experiment of identifying negation cues, improving the results of the dictionary-based system.

Various experiments were conducted on both corpora to compare the performance of both systems under different scenarios; for the task of automatic negation cue detection, we only considered simple negation cue class. These experiments were divided as follows:

- Experiment 1: As a first experiment, the dictionary-based system was applied to detect negation on both T-MexNeg and SFU Review$_{SP}$-NEG corpora.
- Experiment 2: The CRF-based system was trained and tested on the SFU Review$_{SP}$-NEG corpus for the task of negation cue detection. Additionally, the CRF-based system was trained and tested on the T-MexNeg corpus for the task of automatic negation cue detection. The results of these experiments were then compared to generate reference information between both corpora.
- Experiment 3: The CRF-based system was trained and tested on the T-MexNeg corpus to identify only the scope of the negation. Additionally, the CRF-based system was trained and tested on the SFU Review$_{SP}$-NEG corpus to also identify only the scope of the negation. For evaluating the scope identification, we used the predicted negation cues.
- Experiment 4: The CRF-based system was trained and tested on the T-MexNeg corpus to identify only the event of the negation. Additionally, the CRF-based system was trained and tested now on the SFU Review$_{SP}$-NEG corpus to identify only the event of the negation. For evaluating the event identification, we used the predicted negation cues.
- Experiment 5: The CRF-based system was trained and tested on the T-MexNeg corpus to identify negation, scope, and event altogether (Global). Additionally, the CRF-based system was trained and tested on the SFU Review$_{SP}$-NEG corpus to again identify negation, scope, and event altogether. For evaluating the identification of all tags, we used the predicted negation cues.
- Experiment 6: We trained the CRF-based system on the SFU Review$_{SP}$-NEG corpus and tested on the T-MexNeg corpus for the task of negation detection. We then compared the results of this experiment with the results obtained in the first experiment where the CRF-based system was trained and tested on the T-MexNeg corpus for the task of automatic negation detection. This was to compare how the CRF-based system performance differs in different variations of Spanish (Mexican Spanish and European Spanish).
- Experiment 7: As a last experiment, the CRF-based system was trained on the T-MexNeg corpus and tested on SFU Review$_{SP}$-NEG corpus for the task of negation cue detection. We then compared the results with the ones obtained in Experiment 2, where the CRF-based system was trained and tested on the SFU Review$_{SP}$-NEG corpus for the task of negation cue detection.

To ensure that the results are independent of the training and test partitions when training and testing on the same corpus, we used the 10-fold cross-validation technique [52]. Both corpora were split into 10 partitions where each partition or sample was stratified, meaning that each group had the same proportion of tweets with negations as tweets with false negation. In the next section, we present the results for each experiment.

To measure the performance of every experiment, we followed the evaluation methods proposed by Morante and Blanco [42] to implement our evaluation metrics. The performance was measured at sentence level where the system was evaluated on whether every token is correctly labeled or not; to quantify a true positive (TP), it is a requirement that all tokens are correctly predicted; contrariwise, if the system predicted no tag when indeed, no tag was specified in the tested sentence, this results in a true negative (TN). A false positive (FP) occurs when the system predicted a non-existing tag in the tested sentence or when a tag is incorrectly predicted. A false negative (FN) takes place when the system predicted no tag and truly a tag is presented in the tested sentence.

Consequently, standard parameters were calculated to know the ratio of correctly predicted tags to the total predicted tags; precision was computed (TP/TP+FP). To know how many of the actual positives the system captured through labeling them as positive, the recall was calculated (TP/TP+FN). The weighted average of precision and recall was computed through F1-score (2 × (Recall × Precision) / (Recall + Precision)).

## 5. Results

This section shows the results of the experiments performed with the corpus. Table 6 summarizes the results of both dictionary-based system and CRF-based system on the task of negation detection on both corpora T-MexNeg (TMN) and SFU Review$_{SP}$-NEG (SFU)). The CRF-based system was first trained and tested on the T-MexNeg corpus and then trained and tested on the SFU Review$_{SP}$-NEG corpus. As we can see, both systems achieved F1-scores above 85% for each corpus, which is a consequence of the word *no* (no) being the most used negation in both corpora. The negation cue *no* (no) represent 78% of the total negation tags in the T-MexNeg corpus and 61% in the SFU Review$_{SP}$-NEG corpus (see Figure 4). The result is an efficient detection of negation cues, minimizing the number of false positives. Figure 4 shows the 10 most frequent negations in both the SFU Review$_{SP}$-NEG (orange) corpus and the T-MexNeg (blue) corpus.

The CRF-based system gives better results on both corpora, improving the dictionary-based system. F1-score increased from 91% to 95% in the T-MexNeg corpus and from 86% to 88% in the SFU Review$_{SP}$-NEG corpus. This is because the CRF-based system is flexible enough in cues of feature selection, cutting the limitations of the dictionary-based system, which only looks for the words in the dictionary.

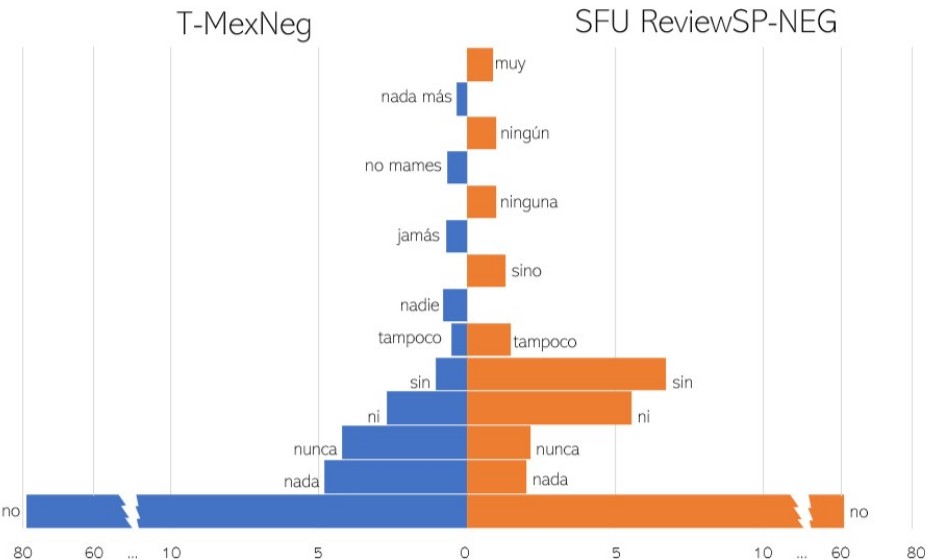

**Figure 4.** Ten most frequent negation in both *T-MexNeg* corpus (represented in blue) and *SFU Review$_{SP}$-NEG* corpus (represented in orange).

**Table 6.** Results for negation cue detection using dictionary and CRF approaches on TNM and SFU corpora.

|  | Experiment 1 Diccionary | | Experiment 2 CRF | |
|  | **TNM** | **SFU** | **TNM** | **SFU** |
|---|---|---|---|---|
| Precision | 0.89 | 0.89 | 0.93 | 0.79 |
| Recall | 0.92 | 0.83 | 0.98 | 0.96 |
| F1 | 0.91 | 0.86 | 0.95 | 0.88 |

Table 7 summarizes the results obtained in the task of identification of the negation cue, the negation event, the negation scope, and all three negation, event, and scope together. The first three columns of Table 7 represent the results when the CRF-based system was trained and tested on the T-MexNeg corpus using 10-fold cross-validation. The last three columns represent the results when the CRF-based system was trained and tested on the SFU Review$_{SP}$-NEG corpus. These experiments aimed to test the same CRF-based system in two different corpora for different tasks (negation, event, and scope).

As we can see, the highest F1 scores on both corpora were obtained in the task of identifying the negation cue. In the task of identification of the event, the F1-score is high for the T-MexNeg corpus (90%) but low for the SFU Review$_{SP}$-NEG corpus (59%). This is because the structure of the negation event in the T-MexNeg corpus is shorter than it is in the SFU Review$_{SP}$-NEG corpus, resulting in a harder task when working on the SFU Review$_{SP}$-NEG corpus.

On the other hand, in the task of identification of the scope, the F1-score reached 75% for the T-MexNeg corpus and 68% for the SFU Review$_{SP}$-NEG corpus. This is a result of the length of the negation scope structure, which is large in both corpora. This complicates the task producing many false positives. Now, for the task of identification of all three negation, event, and scope together (GLOBAL), we obtained the lowest F1-scores out of all the experiments. Since the experiment demands the identification of three different tokens in the same sentence, a high number of errors is conceivable. Besides, considering that a true positive requires that all tokens in a sentence are correctly predicted, these errors will yield false positives resulting in a low Precision.

**Table 7.** Results for negation cue, scope and event identification using the CRF approach on TMN and SFU corpora.

| | | TMN | | | SFU | | |
|:---:|:---:|:---:|:---:|:---:|:---:|:---:|:---:|
| Experiment | Tag | Precision | Recall | F1 | Precision | Recall | F1 |
| 2 | Negation cue | 0.93 | 0.98 | 0.95 | 0.79 | 0.96 | 0.88 |
| 3 | Event | 0.86 | 0.94 | 0.90 | 0.51 | 0.71 | 0.59 |
| 4 | Scope | 0.63 | 0.92 | 0.75 | 0.54 | 0.93 | 0.68 |
| 5 | Global | 0.61 | 0.92 | 0.73 | 0.37 | 0.90 | 0.50 |

In Table 8, we can see the results of the experiment when we trained the CRF-based system for the task of negation cue detection. The column TMN-trained represents the performance scores obtained when training the CRF-based system on the T-MexNeg corpus and testing on the same corpus. The column SFU-trained describes the performance scores obtained by training the CRF-based system on the SFU Review$_{SP}$-NEG corpus and testing on the T-MexNeg corpus. We can see that the F1-score of 0.95 is higher when training on the T-MexNeg corpus and testing on the same corpus using 10-fold cross-validation than when the CRF-based system is trained on the SFU Review$_{SP}$-NEG corpus, which achieved an F1-score of 0.87. These experiments show the importance of a corpus annotated with negation for social media because it can be observed that, when the model is trained on the corpora reviews, the performance metrics (precision, recall, and F1) drops between 6% and 9%.

**Table 8.** Results for negation cue detection testing on the T-MexNeg corpus.

| | Experiment 6 CRF | |
|:---|:---:|:---:|
| | **TMN-Trained** | **SFU-Trained** |
| Precision | 0.93 | 0.87 |
| Recall | 0.98 | 0.89 |
| F1 | 0.95 | 0.87 |

Additionally, Table 9 shows the results of Experiment 7 when the CRF-based system was trained on T-MexNeg corpus and tested on SFU Review$_{SP}$-NEG corpus for the task of negation detection, represented by the column TMN-trained. The results of this experiment are similar to the ones we obtained in the experiment when we trained and tested the CRF-based system on SFU Review$_{SP}$-NEG corpus, represented by the column SFU-trained. When we trained on T-MexNeg corpus, the training data are shorter messages than they are in SFU Review$_{SP}$-NEG; thus, when we test the CRF-based system on SFU Review$_{SP}$-NEG the system does not predict negation when indeed a negation is annotated in the corpus, producing many false negatives that yield a recall of 0.80. On the other hand, when we trained and tested the system on SFU Review$_{SP}$-NEG, tags were incorrectly translated into false positives, resulting in a precision of 0.79. In both experiments, the F1-score achieved is 0.88.

**Table 9.** Results for negation cue detection testing on the SFU Review$_{SP}$-NEG corpus.

| | Experiment 7 | |
| | TMN-Trained | SFU-Trained |
| --- | --- | --- |
| Precision | 0.97 | 0.79 |
| Recall | 0.80 | 0.96 |
| F1 | 0.88 | 0.88 |

## 6. Discussion

As mentioned above, automatic identification of negation is a key task for different Natural Language Processing tasks, for example, sentiment analysis on social media [7]. Our experiments show that a negation identification model trained on reviews (SFU Review$_{SP}$-NEG corpus) presents lower performance metrics when tested on Twitter messages than when the model is trained specifically on Twitter messages. In Table 8, a drop can be observed in all performance metrics when the model is trained on SFU Review$_{SP}$-NEG corpus. For example, the recall drop indicates that this model identifies almost 10% fewer negation cues than the model trained on Twitter data.

In general, the loss of recall on the models trained in a different domain, compared with the performance when using Twitter data is caused by the specific traits existing in the language that is used on social networks. An easy example is the shapes that the simple negation takes on Twitter, going from *no* to *nel, Nelson, Nelson Mandela, Noooo*, etc.

The paper deals with two types of language differences, one provided by the platform (reviews vs. Twitter) and the other related to geographical variation (Spain vs. Mexico). We consider that both categories have features that are crucial for defining a text and expected this to be reflected in the results. Experiment 6 shows how training the model on the T-MexNeg obtains better results for this corpus than training on SFU Review$_{SP}$-NEG. However, Experiment 7 shows that a model trained on the T-MexNeg corpus performed equally to a model trained on the SFU Review$_{SP}$-NEG, when both are tested on SFU ReviewSP-NEG. These results make it difficult to determine the role of the platform and the geographic affiliation in the performance of the system. More precisely, at this stage of the research, we do not have enough data to establish that the dialectal variant is crucial as it was our intuition. More extensive experiments would likely provide more evidence in this line.

Given that social media, in any of its shapes, seems to be a type of communication that is going to last and change the human communication interactions for the future, it is necessary to elaborate resources that can account for the phenomena that are specific to Computer-Mediated Communication. This is, in general, a challenge for NLP, and even more for this key but basic issue of negation identification. This challenge involves not only lexical identification, but also dealing with syntactic creativity, and the pragmatics on the web.

Many of the previous works highlight the importance of negation identification and also use this feature on several tasks for social media [53]. However, most of them tackle the negation identification using rule-based systems due to the lack of annotated corpora [7]. We aim to provide annotated corpora so that the negation identification task in Spanish can be done by machine learning models.

## 7. Conclusions and Future Work

This paper presents the first corpus of Twitter messages in Mexican Spanish with negation annotations. The work explains the process of compilation and tagging of the corpus, as well as the elaboration of the annotation protocol. Additionally, we present experiments on automatic detection of negation terms, events, and scope. We trained a classification algorithm in the TNM corpus and evaluated the model using cross-validation. The obtained results show that the identification of negation terms, event, and scope on Twitter is a challenging task, and it is even harder if the models are trained on corpora of a different domain.

The texts were collected in the geographical area of Mexico, which means that most of them belong to the Mexican variant of Spanish, which is the Spanish dialect with the most speakers in the world [6]. Moreover, they belong to a micro-blogging platform, Twitter. Therefore, we faced the challenge to adapt the annotation to the Mexican variant and the specificity of netspeak.

First, there are different variants in the spellings of words—*nunka, nunk, nunca, nuncA, nuncaaaaaa*, etc. Then, there are the many typographies for capturing pragmatic traits: emojis, capital letters, etc. Moreover, the texts present features such as abbreviations and contractions—*nombre* instead of *no hombre*—and the use of alternative words—*nel, nelson, ño*, or *nop* instead of *no*. Other problems were external references to other tweets that were not in the corpus and Internet slang.

This work deals with syntactic negation, the one built with words that express that something is not occurring through the syntactic structure. In the future, similar resources can be created, covering also lexical negation, the one produced through a morphological change in a word. As an example, *I do not come* can be expressed as *It is impossible for me to come*.

Additionally, the results show how necessary it is to elaborate resources that are explicitly oriented to detect negation in different variants of Spanish.

Finally, another future line of work is building negation corpora based on other social media platforms, such as Facebook, Instagram, or WhatsApp, to better capture the dynamics and idiosyncratic expressions of negation on the web. On this same line, we plan to use previously published corpora that are already accepted by the research community such as those published in TASS@SEPLN (http://tass.sepln.org/tass_data/download.php?auth=QtDa3s5sA4ReWvYeWrf; accessed on 15 February 2021 ).

**Author Contributions:** Conceptualization, G.B.-E. and H.G.-A.; Data curation, S.-L.O.-T. and B.A.-V.; Formal analysis, A.P.; Methodology, G.B.-E., H.G.-A., A.P., S.-L.O.-T. and B.A.-V.; Resources, A.P.; Software, B.A.-V.; Supervision, G.B.-E. and H.G.-A.; Validation, G.B.-E. and H.G.-A.; Visualization, B.A.-V.; Writing—original draft, S.-L.O.-T. ; Writing—review & editing, G.B.-E., H.G.-A. and B.A.-V. All authors have read and agreed to the published version of the manuscript.

**Funding:** This research was funded by CONACyT project CB A1-S-27780, DGAPA-UNAM PAPIIT grants number TA400121 and TA100520.

**Data Availability Statement:** Publicly available datasets were analyzed in this study. This data can be found here: https://gitlab.com/gil.iingen/negation_twitter_mexican_spanish; accessed on 15 February 2021.

**Conflicts of Interest:** The authors declare no conflict of interest.

## Appendix A. Simple Negation Cues Distribution

**Table A1.** Simple negation cues distribution showing words with more than one occurrence.

| Cue | Freq | cue | Freq | Cue | Freq |
|---|---|---|---|---|---|
| no | 3732 | nel | 8 | Nunca nadie | 2 |
| No | 1092 | Jamás | 8 | nunca más | 2 |
| nunca | 177 | Tampoco | 7 | nop | 2 |
| NO | 136 | Sin | 6 | Nooooo | 2 |
| ni | 103 | Nel | 6 | Noooo | 2 |
| sin | 92 | NUNCA | 5 | no no | 5 |
| Nunca | 70 | ningún | 5 | "No, no" | 4 |
| nada | 67 | ni siquiera | 4 | No no no | 2 |
| Nada | 40 | Nooo | 3 | ni tampoco | 2 |
| jamás | 33 | ni de pedo | 3 | Nadie | 2 |
| tampoco | 30 | NI | 3 | Mo | 2 |
| nadie | 10 | Nah | 3 | De nada | 2 |
| Ni | 8 | Ño | 2 | | |

**Table A2.** Simple negation cues with one occurrence.

| | | |
|---|---|---|
| son | SIN | que NOO |
| que NO | puedo | Ñooo no |
| ño | nunk | Nunca Nadie Jamás |
| Nunca de los nunca | Noup | not |
| Nop | NOOOOOOOOOOOO | Noooooooo |
| Noooooooo | nooooooo | Noooooo no |
| Noooooo | noooooo | NOOOOO |
| noooo | nooo | Noo noo no |
| noo | no no y noo | no nooooooo |
| NO NO NOOOOO | No. No. No. | no no no |
| "no, no" | Nombre | nocierto |
| No | Ni puta idea | ninguno |
| Ningún | Ni madres | ni madres |
| Ni de pedo | ni al caso | neeeel |
| Neeee | nadie nunca | nada de |
| Naaaambre | Na | menos |
| JAMÁS | de nada | dejo de pensar |
| Con nada | con nada | |

## Appendix B. Related Negation Cues Distribution

**Table A3.** Related negation cues distribution taking into account words with more than one occurrence.

| Main Cue | Related Cue | Freq | Main Cue | Related Cue | Freq |
|---|---|---|---|---|---|
| no | nada | 135 | nadie | nada | 4 |
| no | ni | 68 | No | nunca | 3 |
| No | nada | 37 | no | ni Obama | 3 |
| no | nadie | 25 | NO | NI | 3 |
| No | ni | 19 | Ni | nada | 3 |
| no | nunca | 13 | ni | nada | 3 |
| no | ningún | 10 | tampoco | nada | 2 |
| sin | nada | 9 | Nunca | ni | 2 |
| no | ninguna | 8 | nunca | nadie | 2 |

**Table A3.** *Cont.*

| Main Cue | Related Cue | Freq | Main Cue | Related Cue | Freq |
|----------|-------------|------|----------|-------------|------|
| No | nadie | 7 | no | no | 2 |
| no | ni nada | 6 | no | ninguno | 2 |
| nunca | nada | 5 | No | ningún | 2 |
| Nunca | nada | 4 | no | nadie más | 2 |
| NO | NADA | 4 | no | nada de nadie | 2 |
| no | NADA | 4 | no | jamás | 2 |
| no | a nadie | 4 | | | |

**Table A4.** Related negation cues with one occurrence.

| | | |
|---|---|---|
| tampoco-ni | Sin-ni nada | Sin-Nancy |
| Sin-nada | SIN-MÁS NADA | NUNCA-NADA |
| nunca-de nadie | nunca-a nada | No-por nada |
| No-para NADA | No-para nada | no-nunk nunkaaaaaaa |
| No-nunca en la vida | no-nuncaaaaa | No-nuncaaaa |
| NO-NUNCA | No-nuncA | no-NUNCA |
| No no-nada | no-ni vrg | No-ni vergas |
| no-ni siquiera | no-ni-ni-ni | NO-NINGUNA |
| No-ninguna | no-NINGUNA | no-ni MERGA |
| no-ni madres | no-Ni | no-nadita |
| no-nadie nunca | no-nadie-ni-nada | No-nadie -nada |
| NO-NADIE | no-nada ni a nadie | no-nada-ni |
| no-nada de NADIE | No-nada...de nada | No-NADA |
| No-Nada | no-nad | no-na |
| No-jamás | no-jamas | no-en lo absoluto |
| no-de nadie | no-de nada | No-a nadie |
| no-a nada | Ni-ni | ni-jamás |
| ni de pedo-si nada | nadie-ni-Ni | nadie-ni-ni |
| nadie-ni | Nadie-nada | nada-nunca |
| Nada-nadie | Mo-nada | jamás-ni-ni |
| jamás-nada | | |

## Appendix C. False Negation Cues Distribution

**Table A5.** Distribution of the negation cues that were considered as false negation in the corpus. The items that appear in the table have more than one occurrence.

| Cue | Freq | Cue | Freq |
|-----|------|-----|------|
| nada más | 39 | no manches | 3 |
| No mames | 35 | no mams | 3 |
| no mames | 18 | no ma | 3 |
| no mamen | 17 | no | 3 |
| No mms | 16 | Ni hablar | 3 |
| de nada | 13 | nada mas | 3 |
| No mamen | 11 | de la nada | 3 |
| Nada más | 11 | y nada | 2 |
| nomas | 8 | Sin duda | 2 |
| No pues | 7 | sin duda | 2 |
| no mms | 7 | si nada | 2 |
| no más | 7 | nunca | 2 |

**Table A5.** *Cont.*

| Cue | Freq | Cue | Freq |
|---|---|---|---|
| No manches | 6 | No que no | 2 |
| nomás | 5 | Nomas | 2 |
| nada | 5 | No mameeeeees | 2 |
| De nada | 5 | no mamar | 2 |
| que nunca | 4 | No ma | 2 |
| pues nada | 4 | no inventes | 2 |
| Para nada | 4 | No bueno | 2 |
| NO MAMES | 4 | no bueno | 2 |
| NO MAMEN | 4 | No. | 2 |
| No mamar | 4 | mas que nunca | 2 |
| más que nunca | 4 | mas que nada | 2 |
| la nada | 4 | Hasta nunca | 2 |
| que no | 3 | como nunca | 2 |
| para nada | 3 | A no | 2 |
| no que no | 3 | a nada | 2 |

**Table A6.** Negation cues that were considered as false negation in the corpus, with only one occurrence.

| | | |
|---|---|---|
| Uuuuuuy no | sou no Yuuna | Sí o No |
| si o no | SINO | Si no |
| sin más | Sin embargo | sin bandera |
| sin | "Si, como no" | reno mames |
| ps nada | por qué no | Poes nada |
| pero no | peor es nada | o no |
| ñoño | nunca vuelvas | nunca jamás |
| Ntp | No te digo | No seas mamón |
| No que pedo | NO QUE NO | no qué |
| No ps | no ps | No pos si |
| No por nada | Nomms nomms | No mmmss |
| NOMBRE | No mas | NO MANCHEN |
| No mancheeen | No mams | No mamn |
| No mammmm | NO MAMESSSSS | No mamesssss |
| no mamessss | No Mamés | Nomames |
| NO MAMEES | no mameees | No mameeeen |
| No mameeeeeen | no mameeeeees | No mamars |
| \\no | No MAMAR | No Ma... ches |
| No maaammmsssss | No maaaaa | No'ma |
| No les digo | No inventes | no es nada |
| No. 24 | No. 12 | No. 1 |
| No | na ma | nmmn |
| ni que nada | ni pez | Ni modo |
| ni modo | NI IDEA | ni hablar |
| \\ nel | Na na na | namas |
| NA' MAAAAS | antes que nada | nada personal |
| nada menos | NADA MÁS | Nada mas |
| Más vale tarde que nunca | más que nadie | más |
| hasta nunca | hasta más no poder | en absoluto |
| don't mames | d la nada | cosas de nada |
| Ay no | ay no | a nada.. a NADA |
| Aaaay no | NO MAMEEEEEEEEEES | |
| Nooooooooooooooooooo no mameeeeeeeeeeeeeeeees | | |

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
