# Peer review of "Negation Detection on Mexican Spanish Tweets: The T-MexNeg Corpus"

_applsci, doi:10.3390/app11093880_

Round 1

Reviewer 1 Report

This work presents a new resource for negation detection in Spanish, specifically for its Mexican variant. In addition, a set of experiments showing its validity is presented. The development of resources to deal with the negation phenomenon is essential since it is crucial for NLP systems and its processing has not been extensively studied. In addition, their annotation is a costly and valuable process. My recommendation is to accept the paper after a major revision. The authors should undertake some important changes that will help to improve the work.

INTRODUCTION

- Please, specify in which language it is annotated the corpus of Retain et al. (line 27).

RELATED WORK

- It would be better to place subsection 1.1 as a new section (Section 2) and include one subsection for English corpora and another one for Spanish corpora.

- Please, include a Table with English corpora annotated with negation as you have done for Spanish.

- In Table 1, you mention the SFU Review SP-NEG corpus. However, you do not include it in the list of Spanish corpora of lines 75-76. Please, include it and the corresponding reference. Moreover, describe it as you have done with the rest of corpora.

[1] Jiménez-Zafra, S. M., Taulé, M., Martín-Valdivia, M. T., Urena-López, L. A., & Martí, M. A. (2018). SFU Review SP-NEG: a Spanish corpus annotated with negation for sentiment analysis. a typology of negation patterns. Language Resources and Evaluation, 52(2), 533-569.

- Regarding Spanish corpora, two corpora that were published in 2020 and 2021 are missing: NUBES and NewsComment.

[2] Lopez, S. L., Perez, N., Cuadros, M., & Rigau, G. (2020, May). NUBes: A Corpus of Negation and Uncertainty in Spanish Clinical Texts. In Proceedings of The 12th Language Resources and Evaluation Conference (pp. 5772-5781).

[3] Taulé, M., Nofre, M., González, M., & Martí, M. (2021). Focus of negation: Its identification in Spanish. Natural Language Engineering, 27(2), 131-152. doi:10.1017/S1351324920000388

- Zafra et al. [22] do not consider the scope among the tags that are typically used to approach negation. They define a method based on a set of syntactic rules that has been proved to be better than the method most used to determine the scope of negation in English tweets.

- Line 102. It is missing [4] a machine learning system to process negation in Spanish. It outperforms state-of-the-art results for negation cue detection and is the first system that performs the task of scope identification for Spanish.

[4] Jiménez-Zafra, S. M., Morante, R., Blanco, E., Valdivia, M. T. M., & Lopez, L. A. U. (2020, May). Detecting negation cues and scopes in Spanish. In Proceedings of The 12th Language Resources and Evaluation Conference (pp. 6902-6911).

- I miss mentioning and referencing the workshops on negation detection for English and Spanish: the BioNLP’09 Shared Task 3 (Kim et al., 2009), the CoNLL2010 shared task (Farkas et al., 2010), the i2b2 NLP Challenge (Uzuner et al., 2011), the *SEM 2012 Shared Task (Morante and Blanco, 2012), the ShARe/CLEF eHealth Evaluation Lab 2014 Task 2 (Mowery et al., 2014),  NEGES 2018 (Jiménez-Zafra et al., 2019a), NEGES 2019 (Jiménez-Zafra et al., 2019b)

CORPUS OF NEGATION IN MEXICAN SPANISH

- What criteria did you follow for collecting the tweets (keywords, hashtags, etc.)? Please, clarify this.

- Line 145. Include an example marking the cue, the scope and the event.

- Line 177. Main negation is not in bold face.

- Line 217. Is the tagging guide available? Please, include the link to it as it could be useful for other researchers.

- I miss a table with the statistics of the corpus: number of tweets (with and without negation), negation tags distribution, number of tokens, vocabulary size, max., min. and avg. number of tokens in a tweet, etc.

- Please, include a table with the agreement between each pair of annotators and the average.

EXPERIMENTS ON AUTOMATIC NEGATION IDENTIFICATION

- Line 265. I miss a reference to the work in [4]. It presents the results of the state-of-the-art for the SFU Review SP-NEG corpus and a machine learning system based on CRF.

[4] Jiménez-Zafra, S. M., Morante, R., Blanco, E., Valdivia, M. T. M., & Lopez, L. A. U. (2020, May). Detecting negation cues and scopes in Spanish. In Proceedings of The 12th Language Resources and Evaluation Conference (pp. 6902-6911).

- I do not understand feature 3 (line 274). You mention four elements (last two, three, four, and five characters) but they are only one feature. How do you represent them? Do you concatenate them?

- On what do you base your selection of the CRF features?

- For scope and event experiment, do you use predicted negation cues or gold cues? You should clarify this.

- For evaluating the systems, do you use the script of Morante and Blanco or do you implement it?

- I miss a final experiment: train on the T-MexNeg corpus and test on the SFU Review SP-NEG corpus.

- In the experiments related to the SFU Review SP-NEG corpus, do you consider the type no-neg for the identification of negation cues?

RESULTS

- In each table of results I would add a first row indicating the experiment to which it refers, e.g. Experiment 1.

DISCUSSION

- Line 458: “Our experiments show that a negation identification model trained on reviews (SFU Review SP -NEG corpus) fails to optimally generalize on Twitter messages.”. This sentence is very harsh, considering that the results are slightly superior and that the language used in both corpora is totally different. 

Typos:

the internet --> the Internet (line 18)

in internet --> on the Internet (line 36)

Trademark --> Treebank (line 76)

Trade Mark --> Treebank (line 77)      

weather --> whether (line 133)

an prepositional --> a prepositional (lines 199-200)

Author Response

This work presents a new resource for negation detection in Spanish, specifically for its Mexican variant. In addition, a set of experiments showing its validity is presented. The development of resources to deal with the negation phenomenon is essential since it is crucial for NLP systems and its processing has not been extensively studied. In addition, their annotation is a costly and valuable process. My recommendation is to accept the paper after a major revision. The authors should undertake some important changes that will help to improve the work.

We thank the reviewer for the time spent reading our work and for their valuable comments. We responded to each of the comments below and made all the suggested changes. We are convinced that with these changes we improved the article.

All the included changes and new additions to the paper are highlighted in blue. Whereas, all the removed sentences are highlighted in red.

INTRODUCTION

- Please, specify in which language it is annotated the corpus of Retain et al. (line 27). 

In line 27, we added the missing information about the language of the corpus: “Reitan et al. [5] built a corpus of negation in English…”

RELATED WORK

- It would be better to place subsection 1.1 as a new section (Section 2) and include one subsection for English corpora and another one for Spanish corpora.

- Please, include a Table with English corpora annotated with negation as you have done for Spanish. 

- In Table 1, you mention the SFU Review SP-NEG corpus. However, you do not include it in the list of Spanish corpora of lines 75-76. Please, include it and the corresponding reference. Moreover, describe it as you have done with the rest of corpora.

Jiménez-Zafra, S. M., Taulé, M., Martín-Valdivia, M. T., Urena-López, L. A., & Martí, M. A. (2018). SFU Review SP-NEG: a Spanish corpus annotated with negation for sentiment analysis. a typology of negation patterns. Language Resources and Evaluation, 52(2), 533-569.

We added a separate Related Work Section, which is now Section 2. In order to make a division between English Corpora and Spanish Corpora we included two subsections. We also included a third subsection related to the automatic methods for negation identification and shared tasks.

We added Table 1 in Section 2.1 with a summary of English Corpora annotated with negation. Two new paragraphs from lines 80 to 107 describe the new table.

From lines 109 to 139, you will find the description of all corpora mentioned in Table 2 (Spanish corpora annotated with negations).

- Regarding Spanish corpora, two corpora that were published in 2020 and 2021 are missing: NUBES and NewsComment. 

[2] Lopez, S. L., Perez, N., Cuadros, M., & Rigau, G. (2020, May). NUBes: A Corpus of Negation and Uncertainty in Spanish Clinical Texts. In Proceedings of The 12th Language Resources and Evaluation Conference (pp. 5772-5781).

We added the missing reference in Table 2 and described it in line 129. “Also the NUBES corpus [25] is in the medical domain with 29,682 sentences from anonymized health records supplied by a Spanish private hospital. It contains 7,567 sentences with negation, annotated with scope, event, and cue

[3] Taulé, M., Nofre, M., González, M., & Martí, M. (2021). Focus of negation: Its identification in Spanish. Natural Language Engineering, 27(2), 131-152. doi:10.1017/S1351324920000388

We added the following sentences from lines 121 to 125: “Similar to the UAM Spanish Treebank corpus [20], it is the NewsCom corpus [24], but instead of news, the corpus consists of 2955 comments posted in response to 18 news articles from online Spanish newspapers.  The news articles cover a variety of nine different topics, two articles per topic.  The corpus consists of 4980 sentences, 2247 marked with negation.  It is annotated with focus, cue, and scope.”

- Zafra et al. [22] do not consider the scope among the tags that are typically used to approach negation. They define a method based on a set of syntactic rules that has been proved to be better than the method most used to determine the scope of negation in English tweets. 

We removed the description of the reference because in that work the authors did not annotate a corpus. As you mentioned here, they implemented a negation detection system by applying several rules.  We still cite this work in the introduction (line 7), as a reinforcement of our claim about the lack of Spanish resources.

- Line 102. It is missing [4] a machine learning system to process negation in Spanish. It outperforms state-of-the-art results for negation cue detection and is the first system that performs the task of scope identification for Spanish. 

[4] Jiménez-Zafra, S. M., Morante, R., Blanco, E., Valdivia, M. T. M., & Lopez, L. A. U. (2020, May). Detecting negation cues and scopes in Spanish. In Proceedings of The 12th Language Resources and Evaluation Conference (pp. 6902-6911). 

We added the reference and include the following sentence in lines 155: Jiménez-Zafra et al. [35] presented a machine learning system based on a Conditional Random Field that processes negation in Spanish, they focused on two tasks: i) negation cue detection and ii) scope identification, outperforming state-of-the-art results for cue detection and being the first system that performs the task of scope detection for Spanish.

- I miss mentioning and referencing the workshops on negation detection for English and Spanish: the BioNLP’09 Shared Task 3 (Kim et al., 2009), the CoNLL2010 shared task (Farkas et al., 2010), the i2b2 NLP Challenge (Uzuner et al., 2011), the *SEM 2012 Shared Task (Morante and Blanco, 2012), the ShARe/CLEF eHealth Evaluation Lab 2014 Task 2 (Mowery et al., 2014),  NEGES 2018 (Jiménez-Zafra et al., 2019a), NEGES 2019 (Jiménez-Zafra et al., 2019b) 

We added the following sentences in lines 175-190 describing workshops and shared tasks in negation detection: “Several shared tasks have emerged to tackle NLP problems regarding negation. BioNLP’09 Shared Task 3 (Kimet al.[41]) centers on the detection of negations and speculation statements concerning extracted events based on the GENIA event corpus of biomedical abstracts written in English. i2b2 NLP Challenge (Uzuneret al.[42]) focused on the automatic extraction of concepts, assertions, negation, and uncertainty on reports written in English provided by the University of Pittsburgh Medical Center. *SEM2012 Shared Task (Morante and Blanco[43]) was dedicated to the identification of the negation, its scope, and focus, two data sets were produced for the task, the CD-SCO corpus of Conan Doyle stories annotated with scopes, and the PB-FOC corpus, which provides focus annotation on top of PropBank, both written in English. ShARe/CLEF eHealth Evaluation Lab 2014 Task 2 (Moweryet al.[44]) focused on facilitating understanding of information in clinical reports by extracting several attributes such as negation, uncertainty, subjects, severity, etc. In this task, the MIMIC II Dataset of clinical reports written in English was used. More recent shared tasks, such as NEGES 2018 (Jiménez-Zafraet al.[45]) and NEGES 2019 (Jiménez-Zafraet al.[46]) aims to advance the study of computational linguistics in Spanish proposing subtasks including annotation guidelines, negation cues detection, sentiment analysis, and the role of negation in sentiment analysis, NEGES uses the SFU ReviewSP-NEG corpus of reviews written in Spanish.”

CORPUS OF NEGATION IN MEXICAN SPANISH

- What criteria did you follow for collecting the tweets (keywords, hashtags, etc.)? Please, clarify this.

We added the clarification of this question in line 193: These tweets are extracted by consuming streaming data from the Standard streaming API. We received every tweet offered by Twitter with the language tag "es"; furthermore, we filtered the tweets by user’s location field in search of the "mx" tag to limit the recollection to the Mexican territory, no other filtering was considered for the extraction process

- Line 145. Include an example marking the cue, the scope, and the event. 

We added Figure 1 with three examples of tweets annotated with the event, cue, and scope. You can find a brief explanation of the first example in line 226: “Figure 1 presents three sentences with negation cue annotated in red, negation event annotated in purple, and negation scope in blue. In the first sentence “@072CDMXEsperemos que le den solución lleva más de 24 horas y no la atienden” (@072CDMX Let’s hope they give you a solution it takes more than 24 hours and they don’t attend her) we can observe the negation cue is the word “no” (don’t), the negation event is the word “la atienden” (attend her), and the negation scope is “no la atienden” (don’t attend her).” 

- Line 177. Main negation is not in bold face.

We added the following sentence in lines 259: “In the next two examples we bold faced only the related negation (Neg\_rel) tag.”

- Line 217. Is the tagging guide available? Please, include the link to it as it could be useful for other researchers.

The tagging guide is available and we also included the link in the paper (line 301): “Based on the description of the above annotation tags, we created an annotation guide that specifies how the tweets should be labeled and under which criteria3

- I miss a table with the statistics of the corpus: number of tweets (with and without negation), negation tags distribution, number of tokens, vocabulary size, max., min. and avg. number of tokens in a tweet, etc.

We included Tables 3 and 4 in Section 3 ( line 312) to show the general statistics for each kind of tweet (with or without negation) so that it can be observed that in both cases, tweets range from 1 to 29 words; tweets that contain negations tend to be almost 3 words longer in average than a regular tweet, however, the standard deviation of the number of tokens is quite high for the length to be a reliable distinctive feature.

- Please, include a table with the agreement between each pair of annotators and the average.

We included Table 5 showing the three pairs of annotators and the agreement for every pair.

EXPERIMENTS ON AUTOMATIC NEGATION IDENTIFICATION

- Line 265. I miss a reference to the work in [4]. It presents the results of the state-of-the-art for the SFU Review SP-NEG corpus and a machine learning system based on CRF.

[4] Jiménez-Zafra, S. M., Morante, R., Blanco, E., Valdivia, M. T. M., & Lopez, L. A. U. (2020, May). Detecting negation cues and scopes in Spanish. In Proceedings of The 12th Language Resources and Evaluation Conference (pp. 6902-6911).

We added the following sentence in line 246: In the work by Jiménez-Zafra et al.[35], a CRF-based system is implemented, they present results of the state-of-the-art for the SFU Review SP NEG corpus where for cue identification the system outperforms state-of-the-art results, while for scope detection they provide the first experimental results. They conclude that the results of the system indicate that the methods used for English are transferable to Spanish.

- I do not understand feature 3 (line 274). You mention four elements (last two, three, four, and five characters) but they are only one feature. How do you represent them? Do you concatenate them?

We corrected the idea (feature 3 is supposed to be 4 different features). We added the following sentences in lines 445-448:

  1. WORD.LastTwoChar: The last two characters of the observed word.                                                5. WORD.LastThreeChar: The last three characters of the observed word.                                                   6. WORD.LastFourChar: The last four characters of the observed word.                                                          7. WORD.LastFiveChar: The last five characters of the observed word.

This has modified the numbering of the whole list.

- On what do you base your selection of the CRF features? 

We added the following sentence in lines 433 - 436: The feature set is inspired by the work done by Loharjaet al. for the shared task of the Workshop NEGES 2018 [30]. They implemented a supervised approach combining CRF and several features for negation cue detection in Spanish for training the model. This approach was ranked in 1st position in the official testing results of the Workshop.

- For scope and event experiment, do you use predicted negation cues or gold cues? You should clarify this.

We evaluated scope and event using predicted cues. We clarify the idea in lines 481-492 where we describe the experiments.

- For evaluating the systems, do you use the script of Morante and Blanco or do you implement it ?

We implemented our own evaluation metric inspired by Morante and Blanco and we added the following sentence in line 509:                                                                                                                                                           To measure the performance on every experiment, we followed the evaluation methods proposed by Morante and Blanco to implement our evaluation metrics.”

- I miss a final experiment: train on the T-MexNeg corpus and test on the SFU Review SP-NEG corpus. 

We added the following sentence in lines 499-502: “Experiment 7: As a last experiment, the CRF-based system was trained on the T-MexNeg corpus and tested on SFU Review SP-NEG corpus for the task of negation cue detection. We then compared the results with the ones obtained in experiment 2, where the CRF-based system was trained and tested on the SFU Review SP-NEG corpus for the task of negation cue detection.

In lines 591-599:  Table 10 shows the results of experiment 7 where the CRF-based system was trained on T-MexNeg corpus and tested on SFU Review SP-NEG corpus for the task of negation detection, the results of these experiments are similar to the ones we obtained in the experiment where we trained and tested the CRF-based system on SFU Review SP-NEG  corpus. When we trained on  T-MexNeg corpus, the training information is shorter messages than they are in SFU Review SP-NEG, so when we test the CRF-based system on SFU Review SP-NEG, null classification occurs producing enough numbers of false negatives to yield a Recall of 0.80. On the other hand, when we trained and tested the system on SFU Review SP-NEG,  tags that were incorrectly predicted translates into false positives, resulting in a Precision of 0.79.In both experiments, the F1-score achieved is 0.88.

- In the experiments related to the SFU Review SP-NEG corpus, do you consider the type no-neg for the identification of negation cues? 

We added the following sentence in lines 469: for the task of automatic negation cue detection, we only considered simple negation cue class

RESULTS

- In each table of results I would add a first row indicating the experiment to which it refers, e.g. Experiment 1. 

We included in all tables indications about the experiments to which the results refer. In some tables, we have results from different experiments so they are identified in each row. You can observe these changes in Tables 5 to 8. In order to make the tables easier to read, we also changed the description of Experiment 1 and Experiment 2 in Section 4.

DISCUSSION

- Line 458: “Our experiments show that a negation identification model trained on reviews (SFU Review SP -NEG corpus) fails to optimally generalize on Twitter messages.”. This sentence is very harsh, considering that the results are slightly superior and that the language used in both corpora is totally different.

We agree with the reviewer and we changed this sentence (now line 655) to: “Our experiments show that a negation identification model trained on reviews (SFU Review SP -NEG corpus) presents lower performance metrics when tested on Twitter messages than when the model is trained specifically on Twitter messages”.

Typos:

the internet --> the Internet (line 18)

in internet --> on the Internet (line 36)

Trademark --> Treebank (line 76) 

Trade Mark --> Treebank (line 77)

weather --> whether (line 133)

an prepositional --> a prepositional (lines 199-200)

Thank you for pointing out these typos, we corrected them all.

Reviewer 2 Report

Comments to authors

The Applied Sciences journal is appropriate for the content of the article. The paper presents a novel resource for treating negation confined to the Mexican dialect used in the social network Twitter. The authors rightly argue that for machine learning models, it is essential that the training corpus is adapted to both the dialect and the genre or textual domain. Therefore, the resource they have developed is justified.

However, the article suffers from serious flaws, both in state of the art and in the experiments' methodology. Therefore, the results of the experiments are not relevant. Furthermore, the style of the article is very repetitive and needs an English professional revision.

The following are the relevant aspects of this review.

Repetitive and careless style:

  • On the same page, the same idea is expressed literally in almost the exact words three times:

Line 29: to the best of our knowledge, there is not an annotated corpus that can help to detect negation on Twitter automatically.

Lines 39-40: However, to the best of our knowledge, there are no Spanish corpora that are annotated with negation in Twitter in Spanish.

Lines 52-53: However, as far as we know, there is no available corpus annotated for the specific case of Mexican Spanish in Twitter data.

  • Errors in the quotation of resources:

Line 76 and 77, Table 1: the UAM Spanish Trademark  the UAM Spanish Treebank

  • Fussy style:

Line 130: “we did not avoid tweets that were answers or retweets”  we did not use retweets in our corpus.

  • Two quotation styles mixed:

Line 264: Agarwal et al. (2010) [29]  Previous work [29] showed that…

  • Wrong terminology:

Line 310: “Spanish from Spain”, better “European Spanish” or  “Castilian Spanish.”

Line 344: “tagger” --> annotator.   “Tagger” is for computational programs, as PoS tagger. “Annotator” is for human experts.

  • Errors:

Table 5: Training with ReviewSp-NEG corpus and testing on T-MexNeg corpus

But in the table, they say TMN-trained   &  SFU-trained.      Very confusing.

  • Again, the idea repeated:

Line 486: As we previously mentioned, automatic identification of negation is a key task for NLP…

  • But the most repeated idea (5 times) is:

“Our corpus consists of 13,707 tweets, … “

In lines:    2, 110, 135, 342  and 481.

  • References are incomplete:

missing pages: references 13, 15, 18, 27, 38, 42

On the background and relevant references

Section 1.1. presents the previous resources (Spanish corpora annotated with negation) to justify the novelty and necessity of T-MexNeg. However, essential references are missing, and the provided description is based on Jiménez-Zafra et al. (2020) [4].

Missing references:

SEPLN (The Spanish NLP Association) organises a Workshop on Semantic Analysis in Twitter (http://tass.sepln.org/) since 2012. All editions have different shared tasks, and for this purpose, they have created several training and evaluation datasets (http://tass.sepln.org/tass_data/download.php?auth=QtDa3s5sA4ReWvYeWrf). Among them is InterTASS, a resource that includes tweets written in Spanish both in Spain and in other American countries (see InterTass v1, InterTASS v2). InterTASS corpus includes a Mexican dataset. It is probably the most important Spanish-language resource on Twitter.

In Table 1,  I suggest sorting it by year and adding two additional columns, availability and link:

Corpus

Year

Domain

Scope

Event

Number of sentences

Annotated Negations

Availability

Link

UAM Spanish Treebank

2013

IXaMed-GS

2015

UHU_HUVR

2017

In line 112, the authors write: “It is important to highlight that none of the previous works on negation detection explore the domain of social media”.

However, one of the most important general reference in the field, Farzindar & Inkpen (2017) NLP for Social Media, said on page 56:

"Many of the methods from the sentiment analysis in the Twitter SemEval task are based on machine learning methods that use a large variety of features, from simple (words) to complex linguistic and sentiment-related features. Mohammad et al. [2013] used SVM classifiers with features such as: n-grams, character n-grams, emoticons, hashtags, capitalisation information, parts of speech, negation features, word clusters, and multiple lexicons.[...] the top system used deep neural networks and word embeddings, and some systems benefited from special weighting of the positive and negative examples. The most important features were those derived from sentiment lexicons. Other important features included bag-of-word features, hashtags, handling of negation, word shape and punctuation features, elongated words, etc.”

(Emphasis is mine).

Another well-known reference is Wiegand et al. 2010, “A survey on the role of negation in sentiment analysis”, in Proc. of the workshop on negation and speculation in NLP, ACL.

The authors should highlight, among the rest, the two most important references on this topic: Jiménez-Zafra et al. (2018) [35] and Jiménez-Zafra et al. (2020) [4].

Also, they mention twice the same reference [15] and [22] in the bibliography section.

Methodology concerns

The authors should follow standard protocols in the field, e.g., Pustejovsky & Stubbs (2017), Natural Language Annotation for Machine Learning: a Guide to Corpus-Building for Applications. The MATTER cycle consists of: 1) Model; 2) Annotate; 3) Train; 4) Test; 5) Evaluate; 6) Revise… and start again in 1).

The first step is “Model the phenomenon.” The authors rely on Martí et al. (2016) but greatly simplify the original model to only three categories. The main categories are Simple negation and Complex negation (recall that I prefer the term used by Martí et al.). The third category is "No negation", which in my opinion is a conceptual error (which Martí et al. also have): they are “False negations” because they are discourse markers, not negative particles. In other words, they should not be included in the corpus annotation but should be identified as negative rules in the annotation guide: “These expressions look like negation but are not negation, do not annotate.”

I miss the annotation guide, which can be published as a link to an open repository, as usual.

The second step is Annotate, and the most important factor is consistency in the way annotators tag the corpus. For assessing how well an annotation task is defined and performed, we use Inter-Annotator Agreement (IAA). In the paper, the authors describe the IIA in the Results section 4.1., not in the Methodology!  They estimate an IAA of 93% overall agreement and 91% of free-marginal kappa agreement. Readers need to check these results in open access to the data.

Besides, the authors describe their criteria for choosing in case of disagreement as follows (lines 356-358): “in the cases where there was disagreement, we were able to search for agreement in two out of three in order to set the repeated label as the definitive one in the final corpus”. This way of proceeding is very unreliable: what if the two matching annotators are wrong. That is why an expert judge (in this case, a professional linguist) is needed to make the final decision to not leave it to mere coincidence. It is the linguist's review that ensures a good Gold Standard (GS).

As Pustejovsky & Amber (2017) explain: “having a high IAA score doesn’t necessarily mean the annotations are correct; it simply means the annotators are all interpreting your instructions consistently in the same way.”

Therefore, without a judge expert, the result of the annotation is neither reliable nor valid.

Similarly, experiments run on this GS assumption are not reliable.

In short, this article has to be based on available open data and comply with standards in the field.

Author Response

The Applied Sciences journal is appropriate for the content of the article. The paper presents a novel resource for treating negation confined to the Mexican dialect used in the social network Twitter. The authors rightly argue that for machine learning models, it is essential that the training corpus is adapted to both the dialect and the genre or textual domain. Therefore, the resource they have developed is justified.

However, the article suffers from serious flaws, both in state of the art and in the experiments' methodology. Therefore, the results of the experiments are not relevant. Furthermore, the style of the article is very repetitive and needs an English professional revision.

We thank the reviewer for their valuable feedback and we address most of their concerns with respect to our methodology and results. We correct the flaws pointed out about the style of the article and the English writing. However, due to time constraints, we were not able to perform a new annotation on a previously published corpus of social media. Although, we plan to do this in future work since we consider it an excellent recommendation. 

All the included changes and new additions to the paper are highlighted in blue. Whereas, all the removed sentences are highlighted in red.

The following are the relevant aspects of this review.

Repetitive and careless style:

  • On the same page, the same idea is expressed literally in almost the exact words three times:

Line 29: to the best of our knowledge, there is not an annotated corpus that can help to detect negation on Twitter automatically.

Lines 39-40: However, to the best of our knowledge, there are no Spanish corpora that are annotated with negation in Twitter in Spanish.

Lines 52-53: However, as far as we know, there is no available corpus annotated for the specific case of Mexican Spanish in Twitter data.

We removed all the repetitive sentences that mentioned the lack of corpora annotated with negations. We only left the first one. 

  • Errors in the quotation of resources:

Line 76 and 77, Table 1: the UAM Spanish Trademark  the UAM Spanish Treebank

We corrected this error

  • Fussy style:

Line 130: “we did not avoid tweets that were answers or retweets”  we did not use retweets in our corpus.

We corrected the style of this sentence: “we also included tweets that were answers or retweets”

  • Two quotation styles mixed:

Line 264: Agarwal et al. (2010) [29]  Previous work [29] showed that…

We uniformed the quotation styles

  • Wrong terminology:

Line 310: “Spanish from Spain”, better “European Spanish” or  “Castilian Spanish.”

Line 344: “tagger” --> annotator.   “Tagger” is for computational programs, as PoS tagger. “Annotator” is for human experts.

We corrected the wrong terminology

  • Errors:

Table 5: Training with ReviewSp-NEG corpus and testing on T-MexNeg corpus

But in the table, they say TMN-trained   &  SFU-trained.  Very confusing.

We corrected the caption of this table.

  • Again, the idea repeated:

Line 486: As we previously mentioned, automatic identification of negation is a key task for NLP…

  • But the most repeated idea (5 times) is:

“Our corpus consists of 13,707 tweets, … “

In lines:    2, 110, 135, 342  and 481.

We removed some of the references to the size of the corpus, but we consider that sometimes it is important to repeat some information so that the reader does not need to go back in the paper. For example, we mention the size of the corpus in the abstract and again when describing it.

  • References are incomplete:

missing pages: references 13, 15, 18, 27, 38, 42

We added the missing pages

 On the background and relevant references

Section 1.1. presents the previous resources (Spanish corpora annotated with negation) to justify the novelty and necessity of T-MexNeg. However, essential references are missing, and the provided description is based on Jiménez-Zafra et al. (2020) [4].

SEPLN (The Spanish NLP Association) organises a Workshop on Semantic Analysis in Twitter (http://tass.sepln.org/) since 2012. All editions have different shared tasks, and for this purpose, they have created several training and evaluation datasets (http://tass.sepln.org/tass_data/download.php?auth=QtDa3s5sA4ReWvYeWrf). Among them is InterTASS, a resource that includes tweets written in Spanish both in Spain and in other American countries (see InterTass v1, InterTASS v2). InterTASS corpus includes a Mexican dataset. It is probably the most important Spanish-language resource on Twitter.

In the Related Work section, we are only mentioning corpora specifically annotated with negations. We acknowledge the relevance of the corpora published by TASS@SEPLN, however, as they are not annotated with negations we did not include them in that section. We added in lines 697 to 699 our intention to use these important resources in future work.

In Table 1,  I suggest sorting it by year and adding two additional columns, availability and link:

Corpus

Year

Domain

Scope

Event

Number of sentences

Annotated Negations

Availability

Link

UAM Spanish Treebank

2013

IXaMed-GS

2015

UHU_HUVR

2017

We added the year to the table as suggested by the reviewer and sorted it correspondingly. However, this paper is not a complete survey on negation identification, so we consider that is not relevant to include the availability and link to the table.

In line 112, the authors write: “It is important to highlight that none of the previous works on negation detection explore the domain of social media”.

However, one of the most important general reference in the field, Farzindar & Inkpen (2017) NLP for Social Media, said on page 56:

"Many of the methods from the sentiment analysis in the Twitter SemEval task are based on machine learning methods that use a large variety of features, from simple (words) to complex linguistic and sentiment-related features. Mohammad et al. [2013] used SVM classifiers with features such as: n-grams, character n-grams, emoticons, hashtags, capitalisation information, parts of speech, negation features, word clusters, and multiple lexicons.[...] the top system used deep neural networks and word embeddings, and some systems benefited from special weighting of the positive and negative examples. The most important features were those derived from sentiment lexicons. Other important features included bag-of-word features, hashtags, handling of negation, word shape and punctuation features, elongated words, etc.”

(Emphasis is mine).

Yes, we agree with the reviewer that many of the previous works highlight the importance of negation identification and also use this feature on several tasks for social media. However, most of them tackle the negation identification using rule-based systems due to the lack of annotated corpora. Our aim is to provide annotated corpora so that the negation identification task in Spanish can be done by machine learning models. (Lines 670-673 of the paper)

Another well-known reference is Wiegand et al. 2010, “A survey on the role of negation in sentiment analysis”, in Proc. of the workshop on negation and speculation in NLP, ACL.

The authors should highlight, among the rest, the two most important references on this topic: Jiménez-Zafra et al. (2018) [35] and Jiménez-Zafra et al. (2020) [4].

We added the following paragraph in lines 169-174: In another vein, some works deal with negation in social media in the context of sentiment analysis. Two key works in this line of research are Mohammadet al.[39] and Wiegandet al.[40]. The former labels with the “_NEG” suffix every word from the negation cue to a punctuation mark and shows how this strategy improves the performance of the system for detecting polarity.  The latter approaches several methods for approaching modeling of negation in sentiment analysis. However, these works are mainly focused on the detection of polarity and do not tackle the structure of negation

However, although these are very important works, we highlight that we are referring to corpora for negation detection. 

Also, they mention twice the same reference [15] and [22] in the bibliography section.

Methodology concerns

The authors should follow standard protocols in the field, e.g., Pustejovsky & Stubbs (2017), Natural Language Annotation for Machine Learning: a Guide to Corpus-Building for Applications. The MATTER cycle consists of: 1) Model; 2) Annotate; 3) Train; 4) Test; 5) Evaluate; 6) Revise… and start again in 1).

We agree in the fact that the method suggested by Pustejovsky & Stubbs is more complete than ours. However, we follow a simplest technique, that is well-known among the computer science community.

The first step is “Model the phenomenon.” The authors rely on Martí et al. (2016) but greatly simplify the original model to only three categories. The main categories are Simple negation and Complex negation (recall that I prefer the term used by Martí et al.). The third category is "No negation", which in my opinion is a conceptual error (which Martí et al. also have): they are “False negations” because they are discourse markers, not negative particles. In other words, they should not be included in the corpus annotation but should be identified as negative rules in the annotation guide: “These expressions look like negation but are not negation, do not annotate.”

We agree in the fact that these are not ‘negations’. We decided to label them as no-negation. This does not have an impact on the automatic detection of negation, but it can be useful for linguistic and statistical purposes.

I miss the annotation guide, which can be published as a link to an open repository, as usual.

The second step is Annotate, and the most important factor is consistency in the way annotators tag the corpus. For assessing how well an annotation task is defined and performed, we use Inter-Annotator Agreement (IAA). In the paper, the authors describe the IIA in the Results section 4.1., not in the Methodology!  They estimate an IAA of 93% overall agreement and 91% of free-marginal kappa agreement. Readers need to check these results in open access to the data.

We moved the IAA to Section 3 and we included the link to the annotation guidelines in lines 301-302 of the paper. The resource is publicly available and it is located on the same platform as the guidelines.

Besides, the authors describe their criteria for choosing in case of disagreement as follows (lines 356-358): “in the cases where there was disagreement, we were able to search for agreement in two out of three in order to set the repeated label as the definitive one in the final corpus”. This way of proceeding is very unreliable: what if the two matching annotators are wrong. That is why an expert judge (in this case, a professional linguist) is needed to make the final decision to not leave it to mere coincidence. It is the linguist's review that ensures a good Gold Standard (GS).

The annotators were all linguistics students and we had a chief linguist who performed the annotation guideline and solved the cases where the annotators disagree. We included this clarification on the paper in lines 304-305.

As Pustejovsky & Amber (2017) explain: “having a high IAA score doesn’t necessarily mean the annotations are correct; it simply means the annotators are all interpreting your instructions consistently in the same way.”

Therefore, without a judge expert, the result of the annotation is neither reliable nor valid.

We agree with the reviewer about the interpretation of the IAA score. We had an expert linguist who reviewed the cases where the annotators disagree.

Similarly, experiments run on this GS assumption are not reliable.

In short, this article has to be based on available open data and comply with standards in the field.

Due to time constraints, we were not able to perform a new annotation on a previously published corpus of social media as the reviewer requested here. Although, we plan to do this in future work since we consider it an excellent recommendation. We included lines 701-703 in the conclusions section.

Reviewer 3 Report

Even though the article is methodogically flawless, it shows at the same time some major defects.

The title suggests to the reader the article would provide some novel results on Mexican Spanish and the special features of the use of tweets in it. However, the connection of the article to the linguistic literature and theory, specifically on the Spanish language and, in this case, cyberpragmatics, remains scarce.

Instead, the article chiefly operates on a methodological a level, in a very specialized setting, and the reader easily gets the impression as if Mexican Spanish itself were not on focus here.

As to the structure of the article, I find it strange that examples of the use of language in tweets are presented in the Discussion section, instead of the one on Results. The Discussion section usually discusses the significance of the findings.

On line 483, the authors write: "the Mexican variant, that is well known by its linguistic creativity." The notion of linguistic creativity is highly subjective: e.g., are there Spanish variants that score lower on the scale of linguistic creativity? It is advisable to leave this out. 

I would recommend the authors shift the focus of the article to a more linguistic direction. The current findings and conclusions should be complemented with a stronger link to the special features of actual Mexican Spanish tweets, from the pragmatic perspective. 

Author Response

Even though the article is methodologically flawless, it shows at the same time some major defects.

All the included changes and new additions to the paper are highlighted in blue. Whereas, all the removed sentences are highlighted in red.

The title suggests to the reader the article would provide some novel results on Mexican Spanish and the special features of the use of tweets in it. However, the connection of the article to the linguistic literature and theory, specifically on the Spanish language and, in this case, cyberpragmatics, remains scarce. 

This is a work that has linguistic interests, but our contribution is building a resource to facilitate the computational approach. The development of a pragmatic theory of negation in social networks is in this moment beyond our possibilities. This is a very interesting line of research for the future, but at this moment, we are centering on the design of experiments. We have highlighted that in lines 43 - 44 of the introduction: Although the main objective of this paper is not to study the language of the Internet, this work can be a solid contribution to the topic, providing data for future studies.

Instead, the article chiefly operates on a methodological a level, in a very specialized setting, and the reader easily gets the impression as if Mexican Spanish itself were not on focus here.

As to the structure of the article, I find it strange that examples of the use of language in tweets are presented in the Discussion section, instead of the one on Results. The Discussion section usually discusses the significance of the findings. 

We have modified the structure of the whole section Discussion.

On line 483, the authors write: "the Mexican variant, that is well known by its linguistic creativity." The notion of linguistic creativity is highly subjective: e.g., are there Spanish variants that score lower on the scale of linguistic creativity? It is advisable to leave this out..

We have modified the paragraph in lines 31-38 of the old version, that developed the same idea. It corresponds to lines 30 - 39 in the new version: 

Additionally, Spanish is a largely spoken language, with native speakers in Europe and America. Mexico has the largest Spanish speaker community, with 123 million in 2020 [6] and 9.5 million users on Twitter. This causes dialect diversification, not only in the lexicon, but also in morphology, syntax, and several slang expressions.  Therefore, the language on the Internet [2], WhatsApp, Twitter, Facebook, etc., shows also great diversity.  Because of this, it is crucial to have a corpus of reference of negation in Mexican Spanish based on social media language

I would recommend the authors shift the focus of the article to a more linguistic direction. The current findings and conclusions should be complemented with a stronger link to the special features of actual Mexican Spanish tweets, from the pragmatic perspective.

We thank the reviewer for their valuable comments. At this point, we do not have the time to change the direction of the paper. We changed the title so it can represent the aim of our research. Lines 670-673 also highlight the aim of the paper:  “Many of the previous work highlight the importance of negation identification and also use this feature on several tasks for social media. However, most of them tackle the negation identification using rule-based systems due to the lack of annotated corpora. Our aim is to provide annotated corpora so that the negation identification task in Spanish can be done by machine learning models.

Round 2

Reviewer 1 Report

The authors have improved the manuscript by considering my comments and those of the other reviewers. However, there are some details they should review. I suggest that the paper be accepted after a minor revision and after answering my comments and the questions I raise.

I think that the previous title “Negation detection on Mexican Spanish Tweets: The T-MexNeg corpus” sounds better than the new one: “Negation detection: The T-MexNeg corpus of Mexican Spanish Tweets”.

SPANISH CORPORA

Lines 113-114. “This corpus is the Spanish version of SFU ReviewSP-NEG Corpus [16]”. The sentence is not correct, it should be: This corpus is the Spanish version of the SFU ReviewEN corpus [16].

METHODS AND SHARED TASK FOR NEGATION IDENTIFICATION

Lines 175-176. Please, describe the BioNLP’09 Shared Task in the past tense as you have done with the rest of shared tasks.

ANNOTATION METHODOLOGY

In Table 3, it is not clear what each variable means. For example, what does mode mean? What do minimum and maximum represent? Please, include the words necessary for a clear understanding of what each variable represents.

In Table 4, what do mean minimum and maximum represent? Minimum of negation tags per sentence? Please, include this information.

Some table titles appear with a period and others do not. Please use the appropriate style for all.

EXPERIMENTS ON AUTOMATIC NEGATION IDENTIFICATION

Line 430. Remove the sentence “They conclude that the results of the system…”. It does not make sense within the description being made.

Line 433. Please, specify that you refer to your feature set. Put something like this: “The feature set used in our study is inspired…”.

Line 436. Here it is also not clear that you are referring to your feature set. You could write something like: “Our feature set includes information about the word…”.

Lines 484, 488 and 492. Please, include the period at the end of sentences.

Table 10. What does the TMN-trained of Table 10 mean? Please, describe it in the previous paragraph.

Please, rewrite the following table captions as they are confusing:

  • Table 7. “F1-score for negation…” is not a good title because you also show other measures. It would be better something like this: Results for negation cue detection using dictionary and CRF approaches on TNM and SFU corpora.
  • Table 8. Idem as Table 7. Suggestion: Results for negation cue, scope and event identification using the CRF approach on TMN and SFU corpora.
  • Table 9 and Table 10 are confusing. Suggestion for Table 9: Results for negation cue detection testing on the TMN corpus. Suggestion for Table 10: Results for negation cue detection testing on the SFU corpus.

Line 596. Why do these null classifications occur? Please, explain this.

Table 9. I am surprised that the results when training and testing with the SFU corpus (experiment 7 - table 10) are inferior to training with the SFU corpus and testing with the TMN corpus (experiment 6 - table 9). Please review the results in table 10 regarding SFU-trained. They should be better since you train and test with the same type of content and language.

DISCUSSION

I miss to mention apart from the differences between the domain of each corpus (reviews and tweets), the differences between the language (Spanish and Mexican Spanish). 

Typos:

  • Line 80. Table 1 summarize ---> Table 1 summarizes
  • Line 102. These tweets are 193 extracted ---> These tweets were extracted
  • Line 228. @072CDMXEsperemos ---> @072CDMX Esperemos
  • Line 230. we can observe the negation cue is the word ---> we can observe that the negation cue is the word
  • Line 230. the negation event is the word ---> the negation event are the words
  • Line 554. the ReviewSP-NEG ---> the SFU ReviewSP-NEG
  • Line 684. which is the Spanish dialect with more speakers in the world. ---> which is the Spanish dialect with the most speakers in the world.

Author Response

The authors have improved the manuscript by considering my comments and those of the other reviewers. However, there are some details they should review. I suggest that the paper be accepted after a minor revision and after answering my comments and the questions I raise.

Thank you for your valuable comments. We reviewed all your comments and performed some new changes to the paper. The new changes can be seen in olive green in the new version of the paper.

I think that the previous title “Negation detection on Mexican Spanish Tweets: The T-MexNeg corpus” sounds better than the new one: “Negation detection: The T-MexNeg corpus of Mexican Spanish Tweets”. 

Thank you for your comments. We will return to the previous title, we also believe that the first title was a better option. 

SPANISH CORPORA

Lines 113-114. “This corpus is the Spanish version of SFU ReviewSP-NEG Corpus [16]”. The sentence is not correct, it should be: This corpus is the Spanish version of the SFU ReviewENcorpus [16]. Luis XXXX

Thank you for the correction, we corrected the reference in line 114 of the paper.

METHODS AND SHARED TASK FOR NEGATION IDENTIFICATION

Lines 175-176. Please, describe the BioNLP’09 Shared Task in the past tense as you have done with the rest of shared tasks. 

We changed the description to past tense (line 177) 

ANNOTATION METHODOLOGY

In Table 3, it is not clear what each variable means. For example, what does mode mean? What do minimum and maximum represent? Please, include the words necessary for a clear understanding of what each variable represents.

The variables in the table were extended to explain what they refer to, for a clearer understanding we simplify the table by removing confusing data. The new description can be seen from lines 316 to 318.

In Table 4, what do mean minimum and maximum represent? Minimum of negation tags per sentence? Please, include this information.

The variables refer to the number of words per tag; we added the clarification in the table’s caption.

 Some table titles appear with a period and others does not. Please use the appropriate style for all. 

We have regularized the captions of tables and figures adding a period if necessary.

EXPERIMENTS ON AUTOMATIC NEGATION IDENTIFICATION

Line 430. Remove the sentence “They conclude that the results of the system…”. It does not make sense within the description being made.

We agree with the reviewer and removed the corresponding sentence (line 440).

Line 433. Please, specify that you refer to your feature set. Put something like this: “The feature set used in our study is inspired…”. 

We modified the sentence in line 443.

Line 436. Here it is also not clear that you are referring to your feature set. You could write something like: “Our feature set includes information about the word…”.

We modified the sentence, now in line 446

Lines 484, 488 and 492. Please, include the period at the end of sentences.

We included the corresponding periods

Table 10. What does the TMN-trained of Table 10 mean? Please, describe it in the previous paragraph. 

We added the following fragments in line 604 and line 606:

“represented by the column TMN-trained”

“represented by the column SFU-trained”

Please, rewrite the following table captions as they are confusing:

  • Table 7. “F1-score for negation…” is not a good title because you also show other measures. It would be better something like this: Results for negation cue detection using dictionary and CRF approaches on TNM and SFU corpora. 
  • Table 8. Idem as Table 7. Suggestion: Results for negation cue, scope and event identification using the CRF approach on TMN and SFU corpora. 
  • Table 9 and Table 10 are confusing. Suggestion for Table 9: Results for negation cue detection testing on the TMN corpus. Suggestion for Table 10: Results for negation cue detection testing on the SFU corpus. 

We rewrote the table captions.

Line 596. Why do these null classifications occur? Please, explain this. 

We modified the sentence and added the clarification of this question in lines 606-612: “When we trained on T-MexNeg corpus, the training information are shorter messages than they are in SFU ReviewSP-NEG, so when we test the CRF-based system on SFU ReviewSP-NEG, the system does not predict negation when indeed a negation is annotated in the corpus, thus producing many false negatives that yield a recall of 0.80. On the other hand, when we trained and tested the system on SFU ReviewSP-NEG, tags that were incorrectly translated into false positives,  resulting in a precision of 0.79. In both experiments, the F1-score achieved is 0.88”

Table 9. I am surprised that the results when training and testing with the SFU corpus (experiment 7 - table 10) are inferior to training with the SFU corpus and testing with the TMN corpus (experiment 6 - table 9). Please review the results in table 10 regarding SFU-trained. They should be better since you train and test with the same type of content and language. 

After a revision of the results, we found a mistake in the information placed in Table 9 (experiment 6) when we trained the CRF-based system on SFU corpus and tested it on TMN corpus. The Precision and Recall metrics are correct but the harmonic mean (f1-score) is not, it is supposed to be 0.87 instead of 0.92. 

P= 0.87, R=0.89   f1= 2*[(P*R)/(P+R)] = 2*[(0.87*0.89)/(0.87+0.89)] = 0.87 

DISCUSSION

I miss to mention apart from the differences between the domain of each corpus (reviews and tweets), the differences between the language (Spanish and Mexican Spanish). 

We added the following paragraph in line 677 of the discussion: 

The paper deals with two types of language differences, one provided by the platform (reviews vs. Twitter), and another one related to geographical variation (Spain vs. Mexico). We consider that both categories have features that are crucial for defining a text, and expected this to be reflected in the results. Experiment 6 shows how training the model on the T-MexNeg obtains better results for this corpus than training on SFU ReviewSP-NEG. However, Experiment 7 shows that a model trained on the T-MexNeg corpus performed equally to a model trained on the SFU ReviewSP-NEG when tested on SFU ReviewSP-NEG. These results make it difficult to determine the role of the platform and the geographic affiliation in the performance of the system. More precisely, at this stage of the research, we do not have enough data to establish that the dialectal variant is crucial as it was our intuition. Probably, more extensive experiments will provide more evidence in this line.

Typos: 

  • Line 80. Table 1 summarize ---> Table 1 summarizes
  • Line 193. These tweets are extracted ---> These tweets were extracted
  • Line 228. @072CDMXEsperemos ---> @072CDMX Esperemos
  • Line 230. we can observe the negation cue is the word ---> we can observe that the negation cue is the word
  • Line 230. the negation event i the word ---> the negation event are the words
  • Line 554. the ReviewSP-NEG ---> the SFU ReviewSP-NEG
  • Line 684. which is the Spanish dialect with more speakers in the world. ---> which is the Spanish dialect with the most speakers in the world 

We corrected these typos.

Reviewer 2 Report

In general, the authors have done a great job in improving the text, not only from the comments of this reviewer but also from some others. I believe that it is in a position to be accepted, although I would like to highlight some aspects that should be modified, as they would help to improve the quality of the paper.

  • Table 1  should be sorted by Domain and then by Year:

Bioscope 2010 Medical
NegDDI-DrugBank 2013 Medical
....
PropBank 2011 Journal
..

Accordingly, the presentation of references (several paragraphs below) should be arranged according to Table 1, as amended.

  • In page 7, "no_negation": I still think this term is poorly chosen. It is a contradiction to use "no" and "negation". It is better to use "false negation" because these cases are not really NEGATIONS themselves, but DISCURSIVE MARKERS which are made up of negative words that have lost their original meaning when grammaticalised. The fact that they follow the nomenclature of Marti et al does not mean that it is correct. 
  • Annotation guidelines should be provided in Spanish AND English.
  • Figure 3 does not have much information, but at least the Normalised Frequency per million or the Relative Frequency should be given, rather than the Absolute Frequency. This way one could compare with the frequencies of those particles in other annotated corpora. This is the standard in Corpus Linguistics.
  • Check bibliographic references. For example, page numbers are written differently. Unify the format. 

Author Response

In general, the authors have done a great job in improving the text, not only from the comments of this reviewer but also from some others. I believe that it is in a position to be accepted, although I would like to highlight some aspects that should be modified, as they would help to improve the quality of the paper.

We thank the reviewer again for their valuable comments. We reviewed all the comments and performed the corresponding changes. The new changes can be seen in olive green in the new version of the paper.

Table 1  should be sorted by Domain and then by Year:

Bioscope 2010 Medical

NegDDI-DrugBank 2013 Medical

....

PropBank 2011 Journal

..

Accordingly, the presentation of references (several paragraphs below) should be arranged according to Table 1, as amended.

We arranged the order of the related work.

In page 7, "no_negation": I still think this term is poorly chosen. It is a contradiction to use "no" and "negation". It is better to use "false negation" because these cases are not really NEGATIONS themselves, but DISCURSIVE MARKERS which are made up of negative words that have lost their original meaning when grammaticalized. The fact that they follow the nomenclature of Marti et al does not mean that it is correct. 

We agree with the idea that ‘no negation’ is not the best term for the phenomenon. We cannot change our tag but have replaced the expression ‘no negation’ with ‘false negation’ throughout the text.

Annotation guidelines should be provided in Spanish AND English. 

Thank you for the reminder, we are currently working on the translation of the annotation guide. We will have it ready by the end of next week.

Figure 3 does not have much information, but at least the Normalised Frequency per million or the Relative Frequency should be given, rather than the Absolute Frequency. This way one could compare with the frequencies of those particles in other annotated corpora. This is the standard in Corpus Linguistics.

We modified Figure 3 by changing the Absolute Frequency to Relative Frequency. 

Check bibliographic references. For example, page numbers are written differently. Unify the format. 

We uniformed the format of the bibliographic references.

Reviewer 3 Report

1. Even though, overall, your statistics is impressive, the reader might still want to see the standard deviations in Table 4, in the light of the means and the minimum and maximum values. You should also comment on the significant variation related to Scope and Event.

2. You should have a native speaker read the text once more before publication: I see some  errors here and there, for example (p. 20): , which is the Spanish dialect with more speakers in the world > most speakers; also, it is pragmatically more correct to use the past tense instead of the present perfect in cases such as (p. 20): The texts have been collected > were collected

Author Response

  1. Even though, overall, your statistics is impressive, the reader might still want to see the standard deviations in Table 4, in the light of the means and the minimum and maximum values. You should also comment on the significant variation related to Scope and Event. 

We thank the reviewer again for their valuable comments. We reviewed all the comments and performed the corresponding changes. The new changes can be seen in olive green in the new version of the paper.

We added the standard deviations in Table 4 and also the comment related to Scope and Event tags. The changes can be seen in olive green from lines 325 to 329:

“Table 4 also shows the frequency of the scope and event tag, where it can be observed that the average number of words within the scope is larger than in the rest of the tags. This variation is expected given that the scope corresponds to all words affected by the negation. Concerning the event tag, it is typically composed of only one word (average), however, there can be more than one word that is specifically negated”

  1. You should have a native speaker read the text once more before publication: I see some  errors here and there, for example (p. 20): , which is the Spanish dialect with more speakers in the world > most speakers; also, it is pragmatically more correct to use the past tense instead of the present perfect in cases such as (p. 20): The texts have been collected > were collected

We corrected the pointed errors. If the paper is accepted we will have a native speaker review the text.